# Conformational dynamics, RNA binding, and phase separation regulate the multifunctionality of rabies virus P protein

Stephen M. Rawlinson [1,12] ✉, Shatabdi Chakraborty [2,3,12], Ashish Sethi [2,3,4,12], Cassandra T. David[1], Angela R. Harrison[5], Lauren E. Bird [1,6], Ashley M. Rozario[7], Sanjeev Uthishtran[8,9], Katie Ardipradja[10], Tianyue Zhao [1], Sibil Oksayan[5], David A. Jans [5], Ching-Seng Ang[3], Zhi Hui Lu[2,3], Fei Yan[2,3], Nicholas A. Williamson [3], Senthil Arumugam [8,9], Vinod Sundaramoorthy[10,11], Toby D. M. Bell[7], Paul R. Gooley [2,3] ✉ & Gregory W. Moseley [1] ✉

RNA viruses encode multifunctional proteins to overcome limited genomic capacity and mediate diverse processes in viral replication and host cell modulation. The rabies virus *P* gene encodes full-length P1 protein and the truncated isoform, P3, which acquires phenotypes absent from P1, including interactions with cellular membrane-less organelles (MLOs) formed by liquid-liquid phase separation (LLPS). This gain-of-function suggests that isoform multifunctionality arises not only from discrete functions of protein modules/ domains, but also from conformational regulation involving interactions of the globular C-terminal domain and N-terminal intrinsically disordered regions (IDRs). The precise mechanisms underlying gain-of-function, however, remain unresolved. Here, we compare the structure and function of P1 and P3, identifying isoform-specific long-range intra-protomer interactions between the IDRs and C-terminal domain that correlate with conformational states, LLPS behavior, and subcellular localization. Mutations in P3 that alter MLO interactions correspondingly modulate these interactions. P1 and P3 can interact with similar/overlapping sets of MLO-associated proteins and have similar LLPS capacity, but only P3 binds RNA, and this interaction correlates with gain-/loss-of-function mutations. Our findings reveal that conformational differences in isoforms regulate LLPS behavior and contribute to protein-RNA interactions, which controls access to host LLPS structures, uncovering a previously unrecognized strategy in P protein multifunctionality.

RNA viruses have limited genomic capacity, but can mediate the infectious cycle, as well as subverting the biology of the infected cell to overcome antiviral responses, hijack the biosynthetic machinery, and otherwise create an environment conducive to replication. Central to this are multifunctional viral genes, which use mechanisms such as RNA editing, alternative reading frames and internal translational start sites to express multiple protein isoforms from limited genomic space[1,2]. Furthermore, the viral proteins expressed are often highly multifunctional, forming extensive interfaces with cellular structures and components[3–6]. Such proteins have typically been considered as simple 'mosaic' assemblies of multiple independent interaction sequences, domains or modules. Thus, mechanisms to add, delete or

alter specific modules between isoforms, such as through protein truncation, are thought to confer functions (through addition of modules) or remove functions (through deletion) to produce a composite phenotype[1,7,8].

However, this simple model cannot account for observed phenotypes, where often quite small viral proteins can target multiple cellular structures and form large interactomes unlikely to be directly accommodated by discrete, specific protein interaction modules[9,10]. Furthermore, the activity of multifunctional proteins often goes beyond the apparent aggregate of protein interaction modules. For example, rabies virus (RABV, a non-segmented negative sense RNA virus of the genus *Lyssavirus*, family *Rhabdoviridae*) has only five genes, which encode structural proteins including the full-length phosphoprotein P1 (residues 1-297, expressed from the P gene). However, the P gene also expresses four accessory proteins (P2-P5) that appear to be produced to mediate processes in immune evasion, and modulation of other host-cell processes[11-13]. P1-P5 have the same sequence, except for the progressive truncation of P2-P5 at the N terminus through translation from in-frame start codons[14]. P1 is an essential cofactor of the viral polymerase (L protein), dependent on an N-terminal sequence including residues 1-19 (required for binding to L protein) (Fig. 1A), and a region in the C-terminal domain (CTD) that interacts with nucleoprotein (N protein) associated with viral genomic RNA[15,16]. The N-terminal region (NTR) also contains a nuclear export sequence (N-NES), which localizes P1 into the cytoplasm[17]. In the cytoplasm, replication occurs within virus-induced 'virus factories' known as Negri bodies, which are membraneless organelles (MLOs) formed by liquid-liquid phase separation (LLPS) and driven by the phase separation properties of P1[18,19]. Of the truncated isoforms, P2 shares a similar cytoplasmic localization to P1, while P3, P4, and P5 are distinct, showing similar phenotypes typically with more nuclear/nucleolar accumulation[11,20]. Among these, P3 is the most abundantly expressed and functionally characterized, and exhibits striking phenotypic differences relative to P1[14].

Previous reports have indicated that truncation to produce P3 (residues 53-297) results in a heterogenous phenotype through interactions and functions that are not evident for P1, including microtubule (MT) association/bundling[11,21], nuclear accumulation[17,20,22], nucleolar targeting[13,23] and association with nuclear bodies (NBs)[24]. Truncation deletes residues 1-52 (Fig. 1A), removing the N-NES and resulting in nuclear localization[17]. This outcome was originally proposed to result from a 'default' phenotype, where loss of the strong N-NES results in dominance of other pre-existing constitutively functional sequences including a C-terminal nuclear localization sequence (C-NLS) (Fig. 1A)[8,25]. However, it was subsequently shown that the truncation results in de novo activation of an N-terminal NLS (N-NLS) comprising a long N-terminal region requiring residues 54-174, that is shared by P3 and P1[20] (Fig. 1A); N-NLS is inactive in P1, with truncation to generate P3 resulting in a direct gain in binding to nuclear import receptors (importins)[20]. Thus, the N-NLS is an extended, non-classical sequence requiring distant residues, which contrasts with typically short NLS sequence motifs, suggesting that conformational organization to form the functional NLS surface is important, and is regulated by truncation.

Other functions/phenotypes reported for P3, but not P1, include association with and bundling of MTs[12,21], and localization to nucleoli and NBs[13,23,24], all of which are dependent on the CTD that is distant from the truncation site. Thus, while N-terminal truncation to generate P3 results in loss of certain functions (including genome replication/transcription and nuclear export), due to deletion/truncation of N-terminal modules, there is also extensive gain-of-function involving downstream, sequentially distant sequence(s) already present but inactive/inhibited in larger isoforms[20,21,23]. This mechanism enhances functionality in the small RABV genome by generating highly distinct protein species from a single coding sequence of the P gene. Together,

these data suggest that P protein functions involve interactions of distant N- and C-terminal regions, such that structural organization, distinct from the functions of individual modules, is critical. Supporting this idea, we recently showed that P3 of a pathogenic RABV strain, Nishigahara, forms distinct open and closed conformational states through interactions between the intrinsically disordered N-terminal regions and the CTD[26]. P3 from an attenuated Nishigahara-derivative strain, Ni-CE, is predominantly open (reduced interactions between N- and C-terminal regions), and defective for characteristic functions such as nuclear localization, MLO interaction and immune evasion, as well as the capacity to undergo LLPS in vitro. This suggested a fundamental mechanism whereby conformational diversity of viral proteins (arising from dynamic structural organization of multimodular proteins) provides an additional level of multifunctionality to augment well understood genetic mechanisms used by viruses to expand genome coding[7,14] (e.g., RNA editing, internal translational start sites, overlapping/overprinted sequences, ribosome slippage/frameshifting, splicing). Based on this hypothesis, broad effects on conformation and intramolecular interactions, rather than local impact of truncation may be a major driver of gain-of-function in viral proteins such as truncated P isoforms, as a key process in viral multifunctionality. However, the nature of the molecular mechanisms and interactions underlying the distinct phenotypes of P1 and P3 are unresolved.

Here, we examine the molecular basis of gain-in-function in P3, finding that the NTR and CTD form extensive interactions within P1 to generate a preferentially closed conformational structure but P3 shows an increase in open states augmenting its conformational diversity. The conformational structures formed correlate with distinct cellular phenotypes including the association of P3 with MLOs, where gain-of-function or loss-of-function mutations correlate with differences in conformational diversity. P1, wild-type (wt) P3 and P3 mutants all undergo LLPS in vitro, and P3 and P1 show overlapping interactions with MLO-associated proteins, indicating that while LLPS, and potentially specific protein associations, are required for MLO-targeting by P3[26], they are not primary determinants in the differing MLO associations of P1 and P3. However, P3, but not P1, binds to RNA, and this is enhanced or lost for gain- or loss-of-function mutants, respectively. Thus, the host interfaces of P protein isoforms appear to involve LLPS and protein-protein interactions as common factors, while protein-RNA interactions are key determinants of isoform-specific interfaces. This identifies the viral protein-host RNA interface as a fundamental mechanism in P protein functional diversity.

## Results

### P3 phenotypes can be specifically modulated by mutation

To investigate the mechanisms of phenotypic differences of P1 and P3, we used proteins of the well-defined RABV strain challenge virus standard-11 (CVS-11), with which MT and nucleolar association, nuclear trafficking mechanisms, and functions at the host interface including in immune evasion were identified[12,13,20,22,23,27,28]. Localization and interactions of P1 and P3 with cellular compartments and structures including the cytoplasm (P1), nucleus, nucleolus, NBs and MTs (P3) have been well-characterized, including P proteins fused to fluorescent protein (for imaging in intact cells to minimize fixation artefacts, of particular importance for MLO analysis[29]) and non-fused proteins in transfected and infected cells[11-13,21-23,25,27,28,30-32]. We thus used analysis of fluorescently fused proteins to assess functions and subcellular localization/interactions including in living cells, with validation of data in fixed, immunostained cells for non-fused and virus-expressed proteins.

To directly compare phenotypic differences between P1 and P3, and mutants of P3, in intact cells, we used proteins fused to GFP for live-cell confocal laser scanning microscopy (CLSM) (Fig. 1B). As previously reported[22-24], GFP-P1 was strongly cytoplasmic, while GFP-P3 could localize into nuclei, nucleoli and additional smaller intranuclear

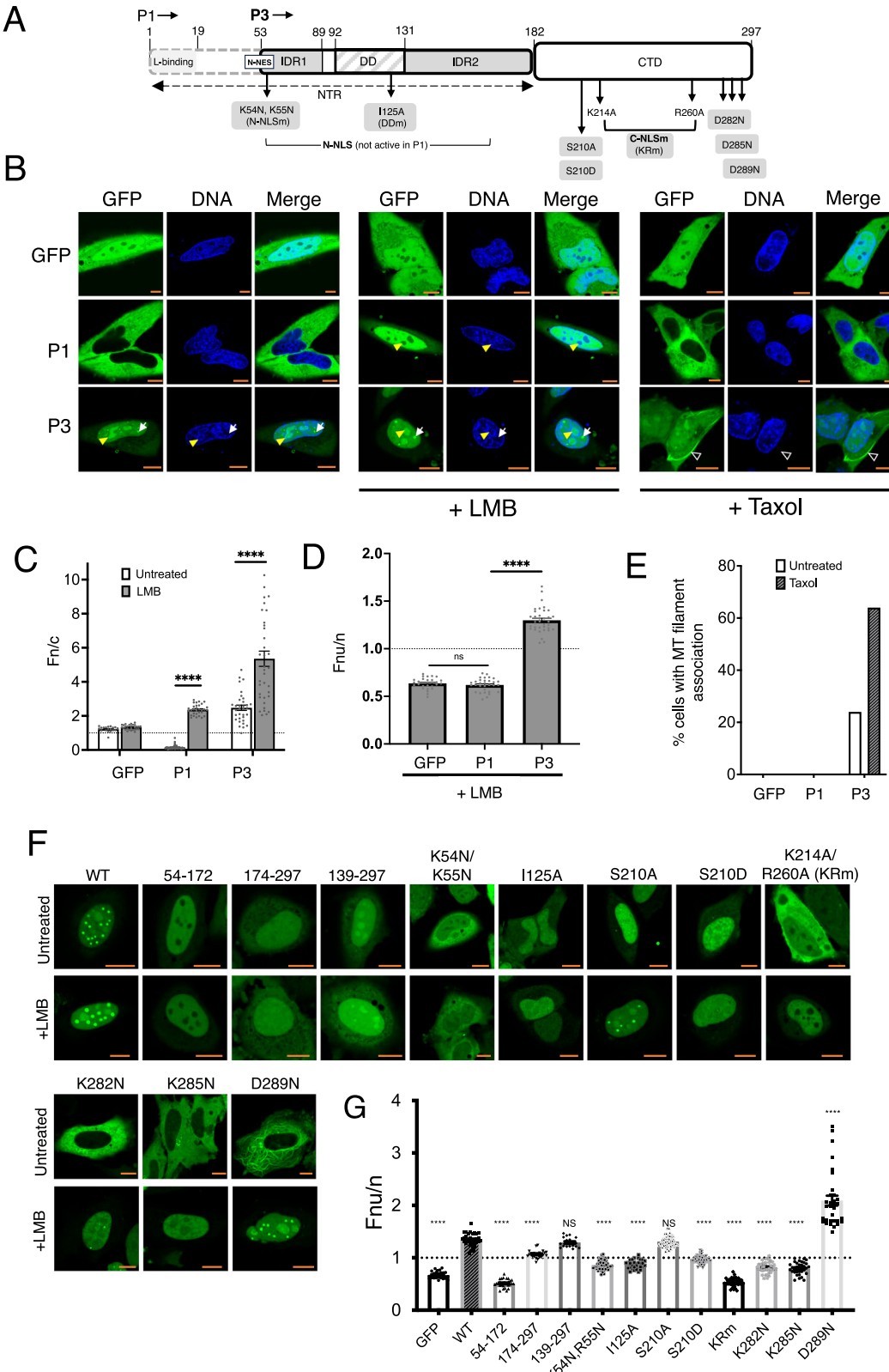

**Fig. legend (panels A–G)**

structures consistent with NBs; we confirmed these phenotypic characteristics using untagged P1 and P3 protein in fixed, immunostained cells (Supplementary Fig. 1). Quantitative image analysis to determine the nuclear to cytoplasmic fluorescence ratio (*Fn/c*) confirmed strong cytoplasmic localization of GFP-P1, with significantly greater nuclear localization of GFP-P3 (Fig. 1C). While GFP-P3 clearly localized to nucleoli, the steady-state cytoplasmic localization of GFP-P1 precludes direct observation of the capacity for nucleolar interaction. However, P1 traffics between the nucleus and cytoplasm due to several NLSs/NESs[17,20] (Fig. 1A) so treatment of cells with leptomycin B (LMB), a specific inhibitor of the exportin CRM1 that mediates P1 nuclear export, results in nuclear localization[17,31] (Fig. 1B, C). Notably, however, P1 remained excluded from nucleoli in LMB-treated cells, with a nucleolar to nuclear fluorescence ratio (*Fnu/n*) comparable to that of

**Fig. 1 | RABV P3, but not P1, associates with multiple cellular MLOs. A** Schematic of the RABV P-protein domain structure, indicating regions of intrinsic disorder (IDR1 and IDR2) and the two structured domains: the dimerization domain (DD) and the C-terminal domain (CTD). The full-length isoform P1 includes the N-terminal residues (1–19) required for binding to the viral polymerase (L protein), which are absent from the N-terminally truncated isoform P3. Key trafficking signals are indicated: N-terminal nuclear localization signal (N-NLS; residues 54–174), N-terminal nuclear export signal (N-NES; residues 49–58), and a conformational C-terminal NLS (C-NLS), which requires residues K214 and R260. Mutations used in this study are shown below the schematic (gray boxes). **B** HeLa cells transfected to express GFP, GFP-P1 or GFP-P3 were treated with or without LMB or Taxol and imaged by live-cell CLSM. Nuclei/DNA were labeled using Hoechst 33342. Yellow arrowheads and white arrows indicate nucleolar and nuclear body (NB) localization, respectively; white unfilled arrowheads indicate MT association. Scale bar = 10 μm (orange), and applies to all microscopy images in the figure. Images such as those in B were used to determine (**C**) Mean nuclear-to-cytoplasmic fluorescence ratio ($F_n/c \pm$ SEM) calculated per cell. Data are from a single experiment where each point represents one cell ($n = 19$ [GFP], 28 [GFP + LMB], 42 [P1], 31 [P1 + LMB], 38 [P3], 41 [P3 + LMB]). Statistical analysis was performed using unpaired two-tailed $t$-tests with Welch's correction due to unequal variances, with $p$ values: P1 vs P1 + LMB,

$p < 0.0001$; P3 vs P3 + LMB, $p < 0.0001$ (****). **D** Mean nucleolar-to-nuclear fluorescence ratio ($F_{nu}/n \pm$ SEM) in LMB-treated cells. Data are from a single experiment where each point represents one cell ($n = 38$ [GFP], 31 [P1], 36 [P3]). Statistical analysis was performed using unpaired two-tailed $t$-tests. Welch's correction was applied for P1 vs P3 due to unequal variances, with $p$ values: GFP vs P1, $p = 0.3082$; P1 vs P3, $p < 0.0001$ (****). **E** Percentage of cells with apparent association of GFP with cytoplasmic filaments consistent with MTs ($n \geq 34$ cells for P1 and P3; $n \geq 19$ cells for GFP alone). **F** HeLa cells transfected to express the indicated truncated P-proteins, or wt or mutant P3 proteins, were treated with or without LMB and imaged live by CLSM. **G** Images such as those in F were used to calculate the mean $F_{nu}/n \pm$ SEM in LMB-treated cells expressing GFP-P3 or the indicated mutants/truncations. Data are from a single experiment where each point represents one cell ($n = 29$ [GFP], 33 [WT], 27 [54–172], 24 [174–297], 18 [139–297], 47 [K54N,R55N], 30 [I125A], 46 [S210A], 40 [S210D], 37 [KRm], 38 [K282N], 34 [K285N], 32 [D289N]). Dotted line indicates $F_{nu}/n = 1$; WT P3 is indicated by dark hatching. Statistical analysis was performed using two-tailed $t$-tests with Welch's correction due to unequal variances. Exact $p$ values (vs WT P3): 139–297, $p = 0.1110$ (ns); S210A, $p = 0.2187$ (ns); all other comparisons, $p < 0.0001$ (****). Source data are provided as a Source Data file.

---

GFP alone, which lacks intrinsic nucleolar localization activity (Fig. 1D). Thus, even if P1 is accumulated in the nuclear compartment, it appears to be incapable of accumulating within nucleoli, consistent with nuclear and nucleolar localization involving distinct mechanisms[33].

Previous analysis has clearly indicated association of GFP-P3 and non-fused P3 with MTs[21]. Furthermore, P3-MT association causes MT bundling, which can be quantified by *d*STORM super-resolution imaging, enabling an indirect method to assess P3-MT interaction in infected cells; analysis using *d*STORM indicated significant MT bundling in cells infected by RABV expressing a wild-type P gene, but not virus expressing the P gene of Ni-CE (that encodes P3 defective for MT bundling), consistent with P3-MT interactions during infection[26]. We confirmed the interaction of P3 with MTs based on the detection by CLSM of filamentous P3-associated cytoplasmic structures that were most pronounced in cells treated with the MT-stabilizing agent Taxol (Fig. 1B, E). The filamentous structures were confirmed to be MTs based on colocalization with β-tubulin in immunostained cells (Supplementary Fig. 2A). No similar structures were observed for P1, with or without Taxol treatment, as previously reported[12,21]. One possible explanation for the lack of detection of MT association by P1 is the presence of high levels of non-MT associated GFP-P1 in the cytoplasm, masking filamentous structures. We thus performed photobleaching to remove the fluorescent signal of soluble, dynamic (i.e., non-MT-associated) GFP-P3 and GFP-P1 (Supplementary Fig. 2B). Photobleaching of P3 resulted in an apparently greater local loss of fluorescence on the bleached side of the cell compared with distant regions, consistent with the protein having a reduced mobility; filamentous fluorescence also remained evident, consistent with reduced cytoplasmic mobility due to interactions with MTs. In contrast, photobleaching of GFP-P1 resulted in an apparently even loss of fluorescence throughout the cytoplasm, with no evidence of residual filamentous protein in photobleached cells, consistent with a lack of MT interaction (Supplementary Fig. 2B); this agrees with data indicating a lack of substantial MT bundling by P1 expression[11] (Fig. 1E). Taken together, these data confirm that P3 gains function in nuclear localization and association with subcellular MLO/LLPS structures (nucleoli, NBs, MTs) compared with P1.

Previously we reported the mutant, K214A/R260A (referred to as KRm[13]), originally identified as impairing nuclear localization involving signals in the CTD[17,22], also impairs nucleolar targeting by P3[13]. Other than this mutant, the molecular details of P3 nucleolar targeting are largely undefined. We thus assessed a panel of additional mutants and truncations of P3 protein for their effect on association with the nucleolus. Other than providing information on nucleolar interaction

specifically, the nucleolus also represents a well characterized LLPS-dependent MLO large enough for routine quantitative analysis by CLSM, and so was selected to provide insight on mutants that may affect LLPS/MLO targeting more generally. Mutations assessed included: I125A, reported to inhibit P protein dimerization[34]; K54N/R55N, inhibits the N-NLS of P3[20]; and S210A and S210D, phosphoinhibitory and phosphomimetic substitutions, respectively, of a protein kinase C phosphorylation site that regulates nuclear import/export[31,35,36]. We also mutated ionizable residues (typically associated with nuclear/nucleolar targeting[33]) in the CTD (Fig. 1A). Truncates comprised the N-terminal region of P3 (P54-172), the CTD (P174-297), and a region including the dynein-light chain association sequence (DLC-AS; residues 143-148)[31] and CTD (P139-297). Several of the mutations affected nuclear import (Fig. 1F, upper panels), so we included LMB-treated cells to analyze nucleolar interactions (Fig. 1F, lower panels).

P54-172 was highly nuclear, and P174-297 localized to the nucleus, with the nuclear localization enhanced by inclusion of the DLC-AS (P139-297; Fig. 1F). This is consistent with the strong N-NLS being present in P54-172[12], and C-NLS in the CTD, the nuclear localization of which is facilitated by the DLC-AS[20]. P174-297 and P139-297, but not P54-172, could localize within nucleoli as expected due to intrinsic nucleolar localization activity in the CTD[23], and P139-297 showed increased accumulation compared to CTD alone (Fig. 1F, G) indicating that the DLC-AS-containing region enhances nucleolar localization.

Of the mutants in the CTD, nuclear accumulation of P3-KRm was reduced compared with wt P3 but enhanced by LMB as previously reported, due to inhibition of interaction of the C-NLS with importins and enhancement of nuclear export that involves a C-terminal NES (C-NES)[17,22] (Fig. 1F). However, nucleolar localization of P3-KRm in LMB-treated and untreated cells remained similar to that of GFP alone, as previously reported[13] (Fig. 1G). P3-K282N and P3-K285N were strongly excluded from the nucleus, but accumulated into the nucleus following LMB treatment; nucleolar accumulation of these proteins following LMB treatment remained significantly reduced ($p < 0.0001$) compared with wt P3, although it was slightly higher than that of P3-KRm or GFP alone. S210A had no significant effect on nucleolar localization in LMB-treated or untreated cells, while S210D was impaired for nucleolar targeting, indicating that phosphorylation may regulate nucleolar interaction. Together, these data further support the role of the CTD in nucleolar localization[13]. In contrast to the inhibitory effect of the above mutations, the CTD mutation P3-D289N, which was excluded from the nucleus in untreated cells, became highly nucleolar following LMB treatment, significantly exceeding the nucleolar localization of wt P3 (Fig. 1F, G).

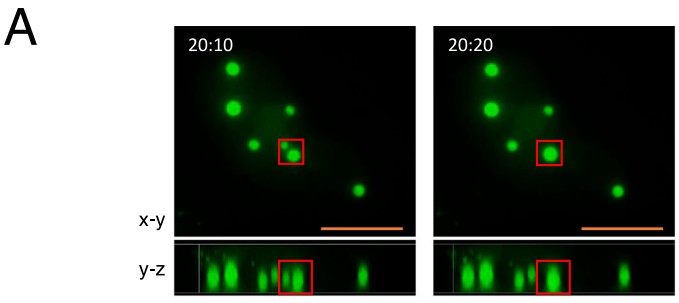

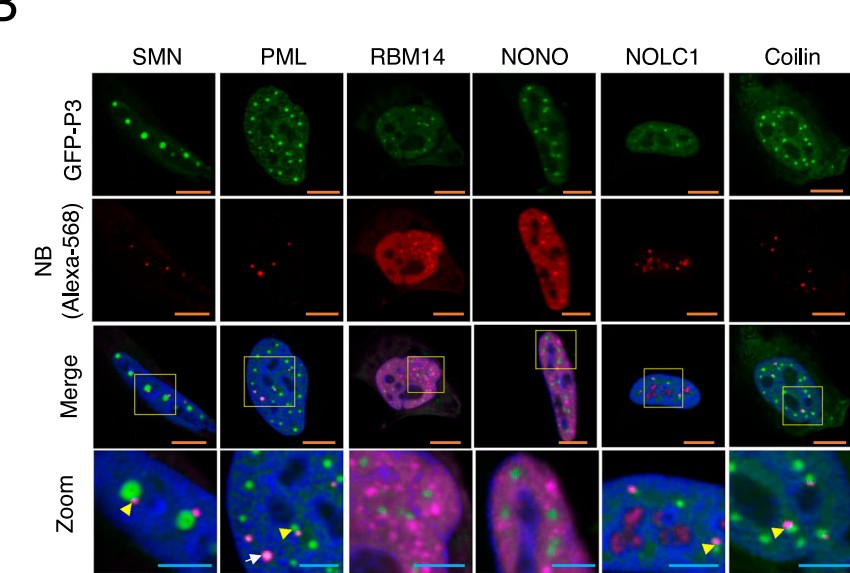

**Fig. 2 | P3 nuclear bodies are liquid and closely associated with known nuclear MLOs. A** P3 nuclear bodies undergo fusion: live-cell timelapse microscopy of HeLa cells transfected to express GFP-P3. Images shown are extracted from Supplementary Movie 1, showing the nucleus of a transfected cell in 3D (x-y and y-z axes are shown) at two separate timepoints (10 s apart). The red box highlights P3 NBs that undergo fusion. **B** HeLa cells transfected to express GFP-P3 were fixed and immunostained with antibodies against protein markers for specific NBs: Cajal bodies (Coilin, NOLC1), paraspeckles (NONO, RBM14), PML-NBs (PML protein); Gem bodies (SMN); and AlexaFluor-568 secondary antibodies (NB (Alexa-568)) before CLSM analysis to image the cell nucleus. White arrows show P3 engulfment by cellular NBs; yellow arrowhead shows cellular NBs proximal to P3 NBs. Scale bars: 10 μm (orange) for main images and 5 μm (light blue) for zoomed panels in (**B**). Merged images (merge and zoom) show the red channel as magenta. Images are representative of ≥5 fields of view.

Of the mutations in the N-terminal region, P3-K54N/R55N reduced nuclear accumulation in untreated and LMB-treated cells, consistent with loss of N-NLS activity[20], and reduced nucleolar localization, indicating that changes at the N-terminus of P3, which has no intrinsic nucleolar localization activity (Fig. 1F and ref. 23) impact on functions mapped to the CTD, consistent with interactions of these domains[26]. Notably, I125A retained nuclear localization, but nucleolar localization was reduced indicating that dimerization may be important. Together, these data support the critical role of the CTD in nucleolar targeting, but also indicate regulatory roles of the N-terminus.

### P3 NBs are liquid-like and associate with nuclear MLOs
P3 interacts with PML protein-containing NBs[24], nucleoli[23], and MT bundles[21]; these interactions have conventionally been considered to arise from protein–protein associations, consistent with reported interactions of P protein with PML protein, nucleolin, and treacle[13,23,24]. However, the lack of apparent interaction of P1 with such structures (Figs. 1B, Supplementary Figure 1, and refs. 23,24,30) suggests that P3 localization involves mechanisms other than canonical domain- or motif-driven protein interactions. Our previous finding[26] that P3 can undergo LLPS in vitro, and that this capacity is impaired by mutations that also disrupt MLO association, indicates that LLPS propensity underpins P3 interactions with cellular liquid compartments/

structures[26]. Thus, P3 may exhibit a general biophysical tendency to associate with nuclear MLOs.

To investigate this, we systematically assessed the localization of nuclear P3 using live-cell CLSM, which avoids artefactual disruption of dynamic, membrane-less NBs that often caused by chemical fixation[29]. As expected, P3 localized to nucleoli and other nuclear puncta[24], which were highly dynamic and frequently underwent contact and fusion events (Fig. 2A, Supplementary Movie 1), consistent with LLPS-driven behavior[37]. Moreover, when nuclear accumulation of P3 was increased by LMB treatment, it appeared to correlate with increased size and number of P3 NBs, consistent with LLPS concentration-dependence (e.g., Fig. 1F); semi-quantitative image analysis clearly indicated that increased nuclear P3 fluorescence ($Fn$) resulted in greater NB formation (Supplementary Fig. 3).

To assess the formation of P3 NBs in infected cells, we used cells infected with CVS-11 RABV. Isoform-specific antibodies are not available, preventing discrimination of P3 from other P protein isoforms that are expressed in infected cells. However, we were able to observe punctate nuclear structures in CVS-11-infected cells immunostained with anti-P protein antibody, which were absent from mock-infected controls (Supplementary Fig. 4). Although the antibody cannot differentiate isoforms, the nuclear structures detected likely reflect P3 (or other shorter isoforms), as the longer P1 and P2 isoforms are largely excluded from the nucleus[25], and P1 will be strongly associated with

Negri bodies[19] (Supplementary Fig. 4). P protein accumulation within nucleoli was not clearly detected in this assay, which may relate to technical limitations, including the known sensitivity of LLPS structures, such as nucleoli, to fixation and immunofluorescence (IF) conditions[29,38]. As inactivation of RABV requires specific fixation conditions, these parameters could not be altered for the assays. However, combined with our previous dSTORM data[26] indicating P3-MT interactions during infection, these findings support that P protein isoforms can form MLO-like nuclear structures during infection, separate from cytoplasmic Negri bodies (Supplementary Fig. 4).

The P3 fusion dynamics observed in transfected cells (Fig. 2A) may relate to prior reports of RABV-induced remodeling of PML NBs[24]. In that study, P3 was found to co-localize with PML protein, and viral infection or P3 expression altered the organization of PML NBs, suggesting that P3 may either target pre-existing PML bodies or recruit PML protein into distinct P3-dependent compartments. To explore this further, we assessed co-localization of P3 NBs with endogenous NBs using IF markers for PML bodies (PML), Cajal bodies (coilin, NOLC1[39]), Gem bodies (SMN), and paraspeckles (RBM14, NONO[40]) (Fig. 2B). Many P3 NBs were clearly distinct from several of these structures, as shown by a lack of co-localization with paraspeckle markers. Other NBs displayed varied spatial relationships with P3 NBs, including partial, incomplete, or adjacent localization (e.g., SMN, PML, Coilin, NOLC1), likely reflecting the dynamic or heterogeneous nature of these structures, including the potential existence of NBs in a pre-fusion or fusion state prior to becoming fully co-localized (Fig. 2B; see also Supplementary Fig. 8B, showing P3 NBs co-localization with NOLC1). Notably, P3 NBs frequently appeared adjacent to Cajal and Gem bodies, which are closely associated with one another in cells[41]. While P3 NBs were often distinct from PML NBs, partial co-localization and apparent engulfment of P3 NBs by PML structures were occasionally observed, consistent with previous findings[24]. These data suggest that P3 and PML-containing NBs can interact or fuse under some conditions, but largely remain distinct entities.

Although the specific roles of P3 NBs remain to be defined, the capacity of P3 to remodel or associate with structures such as PML-NBs[24] suggests a potential role in modulating antiviral responses. While P3 NBs did not co-localize with paraspeckle markers, RABV has been reported to induce the long non-coding RNA, NEAT1[42], the core component of paraspeckles[43], indicating that infection likely effects nuclear reorganization during infection. Together, our results indicate that P3 forms LLPS-driven NBs that selectively associate with a subset of endogenous MLOs. These NBs are dynamic and observable during both transfection and viral infection, supporting roles for P3, through its capacity for phase separation, in nuclear re-organization during infection.

### P3-MT and NB interactions are modified by D289N and KRm

To assess whether mutations affecting nucleolar interactions of P3 also impact on its association with MT bundles[21], we examined CLSM images of cells expressing wt P3, P3-KRm, and P3-D289N (Fig. 3A). P3-D289N associated with extensive cytoplasmic filaments that were readily detectable by CLSM (consistent with bundled MTs) in > 80% of cells (Fig. 3B). In contrast, MT-association was only detected in 23% of cells expressing wt P3, and 4% of cells expressing P3-KRm (Fig. 3A, B).

Using dSTORM super-resolution imaging, we previously showed that expression of P3 of the RABV strain Nishigahara or infection with Nishigahara virus[26] enhances MT bundling, indicated by an increase in MT feature diameter (MTfd)[21]. dSTORM analysis confirmed that the enhanced GFP-P3-MT association observed in cells expressing P3-D289N correlates with increased MTfd compared with cells expressing wt P3, while the MTfd was reduced in cells expressing P3-KRm (Fig. 3E, F). Wide-field images of cells used for dSTORM indicated that the β-tubulin-immunostained MT filaments analyzed colocalize with wt and D289N P3-associated filaments, but little to no equivalent

colocalization was evident for P3-KRm, confirming the differing phenotypes of these mutants corresponds to differing MT interactions (Supplementary Fig. 5). Thus, the effects of KRm and D289N on the capacity of P3 to interact with and bundle MTs correlates with their effects on the nucleolar association. MT bundling is dependent on LLPS[44], indicating that P3 may induce LLPS at associated MTs (similar to the cellular MT-associated protein, Tau), or facilitate phase separation of MTs mediated by cellular mechanisms[44–46].

MT-association and bundling is associated with P3-mediated antagonism of signal transducers and activators of transcription (STAT)1/STAT2-dependent antiviral interferon (IFN) signaling[12,21]. To determine whether altered MT association/bundling correlates with altered STAT1/2 antagonism, we assessed the IFN-antagonist function of wt P3, P3-KRm and P3-D289N using a dual luciferase reporter assay, as previously described[21]. All of the P3 proteins displayed IFN-antagonist function compared with controls (P3 truncated for 30 residues of the CTD [Δ30], or GFP alone, which lack STAT1-targeting activity[28]), but the activity of P3-D289N and P3-KRm was enhanced and impaired, respectively, compared with wt P3 (Fig. 3G). P3-KRm mutation does not prevent binding to STAT1[28], consistent with mutations having effects on antagonistic mechanisms (i.e., MT association/bundling) rather than disabling P protein-STAT1 interaction. Taken together with previous observations that P3 from the pathogenic RABV strain Nishigahara induces strong MT bundling and has enhanced STAT1 antagonist activity compared to P3 from the attenuated Nishigahara-derivative strain Ni-CE (which is impaired in MT bundling), these findings support a role for MT association and bundling in P3-mediated IFN antagonism[21].

Notably, mutants affecting nucleolar association also appeared to affect NB association (Fig. 1F). Quantitation of P3-expressing cells based on apparent P3-NB interaction indicated that wt P3 and P3-D289N (in cells treated with LMB to enable nuclear localization) formed NBs in a substantial proportion of cells, but that this was strongly impaired for P3-KRm (Fig. 3C). P3-D289N did not enhance NB association compared with wt P3 (and in fact appeared to reduce association) (Fig. 3C), but this is likely due to the increased localization of P3-D289N into the more prominent nucleoli (Fig. 3A, D), sequestering P3 from the nucleoplasm. The concordant influence of mutations on nucleolar, NB and MT-bundle association suggest a common effect on the interactions of P3 with different MLOs.

### P1 and P3 share interactors post-lysis but differ in cells

P protein forms multiple interactions with cellular proteins including importins[22], innate immune signaling proteins[47], focal adhesion kinase (FAK)[48], ribosomal protein L9 (RPL9)[49], PML proteins[24] and the nucleolar proteins NCL[23] and treacle[13], many of which associate with MLOs structures[50]. Interaction of proteins with these structures is usually associated with an ability of the proteins to undergo LLPS, involving IDRs, and protein-protein and protein-RNA interactions[50,51].

To assess the interactions of P3 and P1 with MLO-resident proteins, we expressed GFP, GFP-P3 or GFP-P1 in HEK-293T cells before immunoprecipitation (IP) using GFP-Trap, and analysis by mass-spectrometry (IP-MS) (see Supplementary Fig. 6 for principal component analysis (PCA); and Supplementary Data 1 for complete list of proteins identified). 410 proteins were significantly enriched ($p < 0.05$) in GFP-P3-expressing cells compared with GFP alone, with 249 proteins identified as high-confidence interactors ($p < 0.01$) (Fig. 4A). The identification of NCL, a validated P3 interactor that is important to RABV infection[23], as well as other validated P protein interactors such as STAT1[27], RPL9[49] and importin-α2[22] reinforced confidence in the dataset. Gene ontology (GO) analysis of the P3 interactors suggest a strong association with ribosomal functions, translation and RNA binding (Fig. 4C), consistent with the localization of P3 to nucleoli and associated role in inhibiting rRNA biogenesis[13]. Notably, over one-third of the interactors identified are proteins associated with one or more

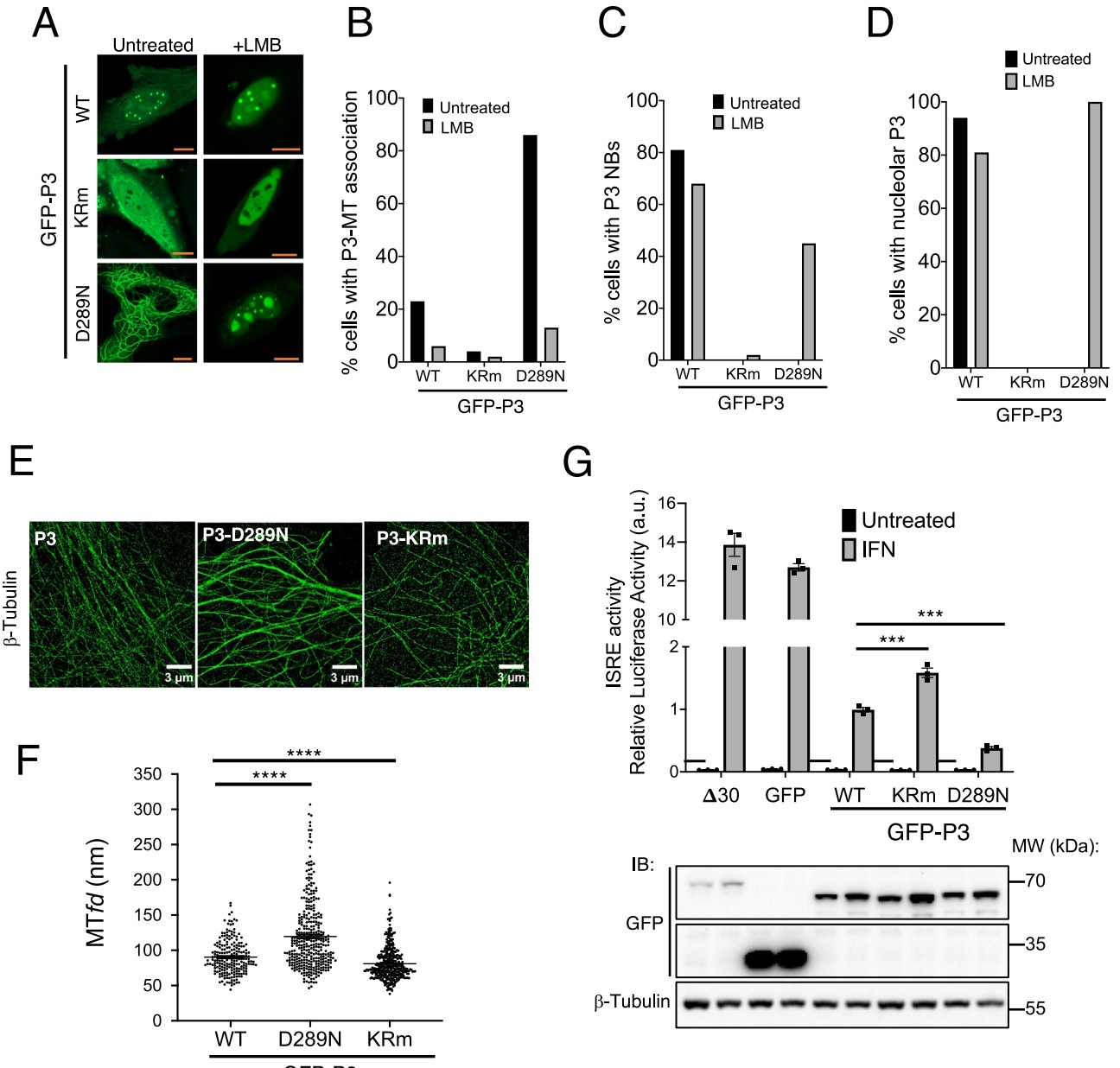

**Fig. 3 | P3 mutants KRm and D289N impair and enhance, respectively, association with cellular MLOs and IFN antagonist function. A** HeLa cells transfected to express the indicated GFP-tagged proteins and treated without or with LMB were imaged live by CLSM. Images shown are representative of ≥2 independent assays with ≥11 fields of view analysed per experiment; results are consistent with observations from multiple additional assays. Scale bars = 10 μm (orange). **B–D** Images such as those in A were analysed to determine the percentage of cells with (**B**) cytoplasmic filamentous P3 (MT-association), (**C**) P3-NB association, (**D**) nucleolar accumulation of P3 (i.e., levels of GFP-P3 fluorescence higher in the nucleoli than in the nucleoplasm). **E** direct Stochastic Optical Reconstruction Microscopy (dSTORM) images of immunostained β-tubulin in COS-7 cells expressing the indicated proteins (see Supplementary Fig. 5 for epifluorescence images of cells used for dSTORM analysis). **F** Microtubule feature diameter (MTfd) quantified from images such as those in (**E**) and presented as scatter dot plots. Each point

represents a single MT ($n = 212$ [wt P3], 348 [P3-D289N], 391 [P3-KRm]) combined data from two independent experiments. Statistical analysis was performed using unpaired two-tailed t-tests with Welch's correction due to unequal variances. p-values for both wt P3 vs P3-D289N and wt P3 vs P3-KRm are $p < 0.0001$ (****). **G** IFN-dependent signaling was assessed using HEK-293T cells expressing the indicated proteins using an ISRE-dependent dual luciferase reporter assay, as previously[17,29]. Data show mean ± SEM from three biological replicates, representative of two independent experiments. Statistical analysis was performed using unpaired two-tailed t-tests with Welch's correction for unequal variances. Exact p values: WT vs KRm, $p = 0.0020$ (***); WT vs D289N, $p = 0.0019$ (****). Immunoblots of lysates used in luciferase assay, probed with antibodies to GFP and β-tubulin (lower panel). Molecular weight (MW) markers were run on the same gels and marker positions are indicated. Source data are provided as a Source Data file.

cellular MLOs, including nucleoli (e.g., NCL, nucleophosmin (NPM1), fibrillarin (FBL), RNA helicase DDX21, and multiple ribosomal proteins), stress granules (SGs) (e.g., G3BP1, G3BP2, PABPC1), P-bodies (e.g., ZC3HAV1) and nuclear bodies such as paraspeckles, Cajal bodies and nuclear speckles (e.g., NOLC1, RBM14, PHF5A) (Supplementary Fig. 7A).

377 proteins were enriched in P1 compared to GFP IPs ($p < 0.05$) with 206 high-confidence proteins ($p < 0.01$); interestingly, despite the apparent lack of association of P1 with MLOs in cells, many of the interactors identified for P1 were also identified for P3 (Fig. 4B). Direct quantitative analysis to compare GFP-P1 and GFP-P3 confirmed there are few significant differences between the interactions formed by P1

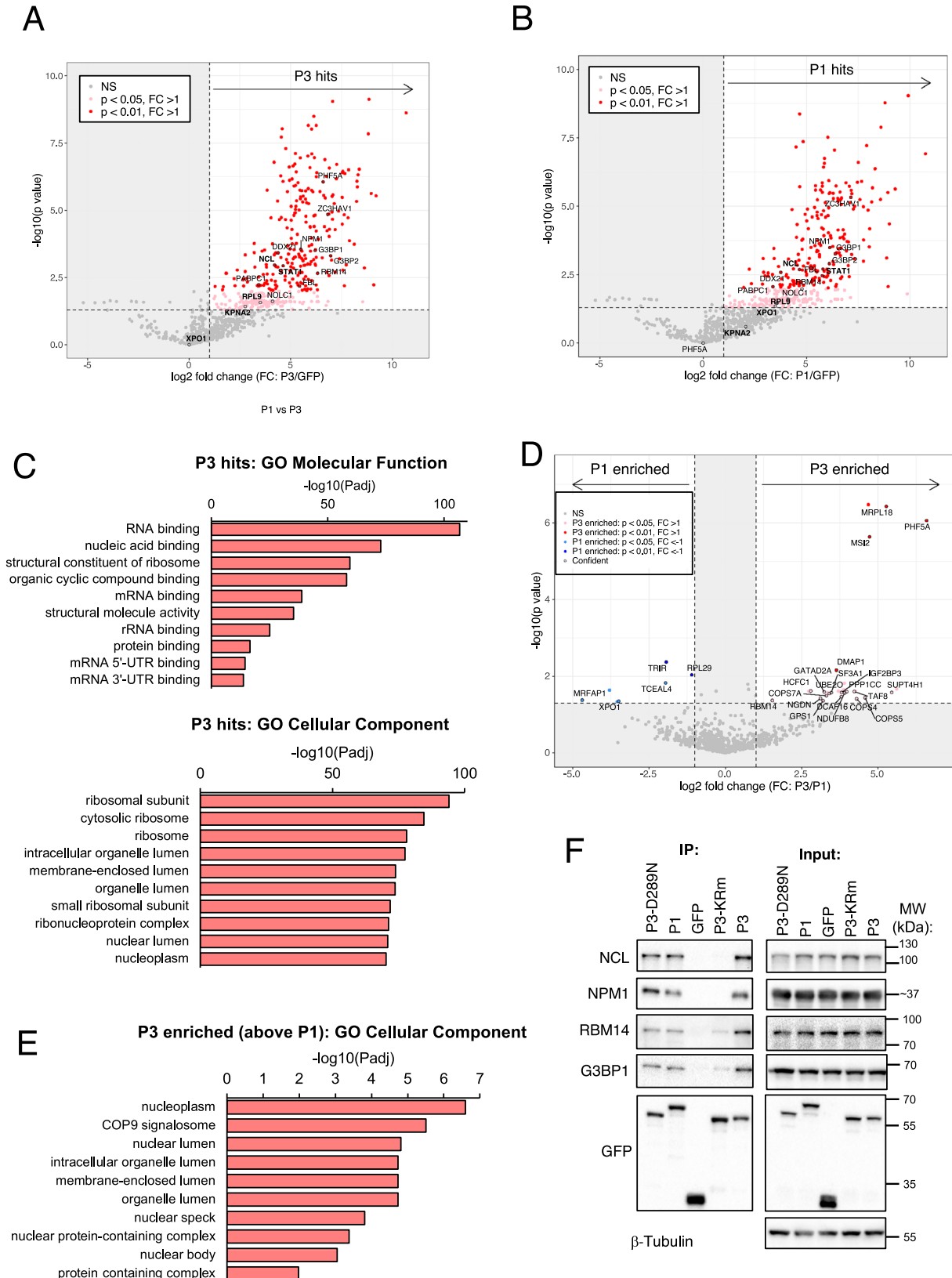

compared with P3 (Fig. 4D), with only five proteins identified as significantly enriched for P1 and 20 proteins for P3. Among these, the nuclear export factor XPO1/CRM1 was enriched for P1, consistent with the strong nuclear export activity of P1 that is mediated by the XPO1/CRM1 dependent N-NES[17]. GO analysis of interactors enriched in P3 IPs indicated proteins associated with nuclei and NBs (Fig. 4E).

To validate some of the main MLO-associated interactors of P3 and P1 we analyzed IPs by immunoblotting (IB) for selected proteins associated with nucleoli (NCL, NPM1), paraspeckles (RBM14), nuclear speckles (NPM1, RBM14), and SGs (G3BP1). All of the proteins tested co-precipitated with wt P3 and P1 but not GFP alone (Fig. 4F), correlating with our quantitative IP-MS data (Supplementary Fig. 7B).

**Fig. 4 | P1 and P3 bind proteins associated with multiple MLOs. A, B** Volcano plots showing interactors ('hits') of (**A**) P3 and (**B**) P1 identified by IP-MS that were enriched above GFP control ($p$ value < 0.05 and log2 fold change (FC) > 1). Red dots indicate high-confidence interactors, faded red dots indicate lower-confidence interactors, and gray dots represent proteins not significantly enriched. Selected proteins of interest are labeled and circled in black. Statistical significance was assessed using two-sided, unpaired $t$-tests in Perseus on $\log_2$-transformed LFQ intensities, with missing values imputed. Hits were considered significant at $p < 0.05$ and $\log_2$ fold change > 1. **C** Functional annotation enrichment of P3 interactors. **D** Volcano plot showing P protein interactors enriched in P1 (left side; $p$ value < 0.05 and log2 FC < −1) or P3 (right side; $p$ value < 0.05 and log2 FC > 1). "Confident" interactors were also enriched above GFP control and are circled in black. **E** Functional annotation enrichment of P3-enriched interactors. **F** IPs of the indicated P3 proteins or GFP alone (control) were analysed by IB using the indicated antibodies. Molecular weight markers were run on all gels; marker positions are indicated, and for NPM1 the approximate size ( ~ 37 kDa) is shown as the marker lane was cropped. Uncropped blots with markers are provided in the Source Data file. Source data are provided as a Source Data file. Mass spectrometry source data are available via ProteomeXchange with identifier PXD064752.

Interaction were also clearly detected for P3-D289N, but were strongly reduced or not detectable for P3-KRm. Thus, it appears that despite differences in MLO localization in intact cells (e.g., Fig. 1), P1 and P3 share conserved interaction sites, enabling co-precipitation with proteins after solubilization, and that changes to structure and/or effects on interaction sites(s) in P3-KRm result in loss of interaction with MLOs and with MLO-associated proteins.

Notably, previous studies on P3-NCL interactions reported that NCL co-precipitates with both P3 and P1 in IPs, even though P1 and NCL do not co-localize in cells, where P1 is cytoplasmic and NCL is primarily nuclear/nucleolar[23]. To assess whether there is similar colocalization of P3, but not P1, with nuclear interactors identified in our IP-MS, we selected NPM1 and NOLC1 for analysis using IF. This revealed co-localization of both NPM1 and NOLC1 with P3 in nucleoli, and of NOLC1 with P3 in NOLC1-associated NBs, which are likely Cajal bodies (Supplementary Fig. 8A, B; and Fig. 2B). However, P1 did not co-localize with either protein due to its cytoplasmic localization and lack of any apparent re-localization of the cellular proteins. In contrast, colocalization of the newly identified interactor G3BP1, a cytoplasmic protein that is essential for SG formation, was observed for both P1 and P3 in the cytoplasm of some cells (Supplementary Fig. 8C). Colocalization with P3 was most evident in a subset of cells that exhibiting SG formation (evidenced by cytoplasmic G3BP1 puncta), where SGs frequently colocalized with cytoplasmic regions enriched in P3; similar localization was apparent for P1, but this was more variable, with colocalization only evident in a subset of SG-positive cells. These data indicate a level of association between G3BP1 and P3, and possibly P1, in the cellular context (Supplementary Fig. 8C). Together, these data suggest that, while P1 can form interactions with many MLO proteins in IP assays using solubilized cells (in which cellular compartmentalization is disrupted), such interactions do not necessarily occur in a native cellular context. Importantly, this lack of interaction is not merely due to conventional compartmentalization between membranous organelles, as even when P1 is enriched in the nucleus following LMB treatment, it fails to accumulate and co-localize with interactors in nucleoli (Fig. 1B).

A limitation of the use of transfection/overexpression-based approaches particularly relevant in the context of LLPS-associated phenomena, is the potential effects on physiological protein stoichiometry and localization. Experiments to analyze specific P protein isoforms in the context of viral infection is limited by the lack of isoform-specific antibodies, as well as the requirement for fixation which can impact on subcellular organization/localization, particularly of LLPS structures (above). Nonetheless, multiple previous data and the data in this study (Supplementary Fig. 4) support the biological relevance of our findings, including alignment of our observations of P3-LLPS-associated phenotypes with data from infection-based assays on interactions (i.e., P protein NB formation (Supplementary Fig. 4), MT bundling[26], effects of infection on PML NBs[24]) and functional roles of identified P1/P3 interactors during viral infection (e.g., NCL[23], RPL9[49], Staufen-1[52]). Together, these data support the idea that LLPS of RABV proteins has functional relevance independent of replication/Negri bodies. Importantly, our results suggest that while P1 and P3 share an ability to bind numerous MLO-associated cellular proteins,

their interactions in intact cells likely differ substantially due to differing localization/association with MLOs. Moreover, since P1 and wt and mutant P3 proteins retain LLPS activity in vitro (see below), differences in MLO association do not appear to be primarily dictated by this property or by protein-protein interactions alone.

## P1 and P3 form distinct structures

Recent analysis of P3 protein from the Nishigahara and Ni-CE strains of RABV suggest that broad structural organization of modules, through interactions of the P3 C- and N-terminal regions, are important to specific P3 functions, such that P3 may form several conformations acting as distinct functional protein species[26]. Such conformational diversity could explain the unique functions of P3 compared with P1 (which includes the complete sequence of P3), providing a protein-based mechanism to augment the genetic mechanisms used by viruses to enhance functionality[7,14].

To compare the structures of P1 and P3, we recombinantly expressed and purified the CVS-11 proteins for Small-Angle X-ray Scattering (SAXS) and cross-linking Mass Spectrometry (XL-MS) analysis. P1 comprises two structured domains, a helical dimerization domain (DD), a globular CTD, and two IDRs (IDR1 at the N terminus and IDR2 between the DD and CTD) (Fig. 1A). SAXS analysis generated a profile of P1 consistent with the expected dimeric protomer, with a mass of ~88 kDa (expected mass 67 kDa) showing a radius of gyration ($R_g$) of 49.25 Å and maximum intramolecular dimension ($D_{max}$) of 170.11 Å (Fig. 5A, and Supplementary Fig. 9, Supplementary Table 1). These data are consistent with a protomer comprised of a significant content of disorder as depicted in a Kratky plot (Fig. 5E). To characterize the structure and the conformational diversity within the disordered regions of P1, we used the Ensemble Optimization Method (EOM)[53,54] to generate a pool of different possible conformations of IDRs with respect to the ordered domains. A significant portion ( > 60%) of the conformations consistently highlighted spatial proximity between the ordered domains and the disordered regions in P1, forming closed or compact states (Fig. 5G), in agreement with previous SAXS and NMR analysis of P1, albeit from a different strain of RABV[55]. A SAXS analysis of wt P3 was also consistent with a dimeric protomer comprised of regions of order and disorder, of apparent mass of ~63 kDa (expected mass 56 kDa), $R_g$ of 41.60 Å and $D_{max}$ of 144.38 Å (Fig. 5D, F; and Supplementary Fig. 9; Supplementary Table 1). EOM (Fig. 5H) analysis of P3 also showed a broad and heterogeneous pool of conformations, similar to that which we previously observed for P3 of the Nishigahara strain[26], and highlighting the fluctuation between closed and open states where, in the closed states, the IDRs preserved an interaction with the ordered domains, but to a lesser extent than P1, likely due to the shorter IDR1.

To further investigate and compare the structural topology and predicted proximity of the domains in P1 and P3, we used XL-MS with disuccinimidyl sulfoxide (DSSO) crosslinker, which covalently links the sidechain amine groups of lysine residues that are between 10 and 30 Å apart[56–58]. These XL-MS data support the SAXS analyses that indicate that the CTD, IDR1 and IDR2 interact with each other in both P1 and P3 (Supplementary Data 2). However, P1 showed eight additional crosslinks not observed in P3 (Fig. 6A; Supplementary Data 2): two between

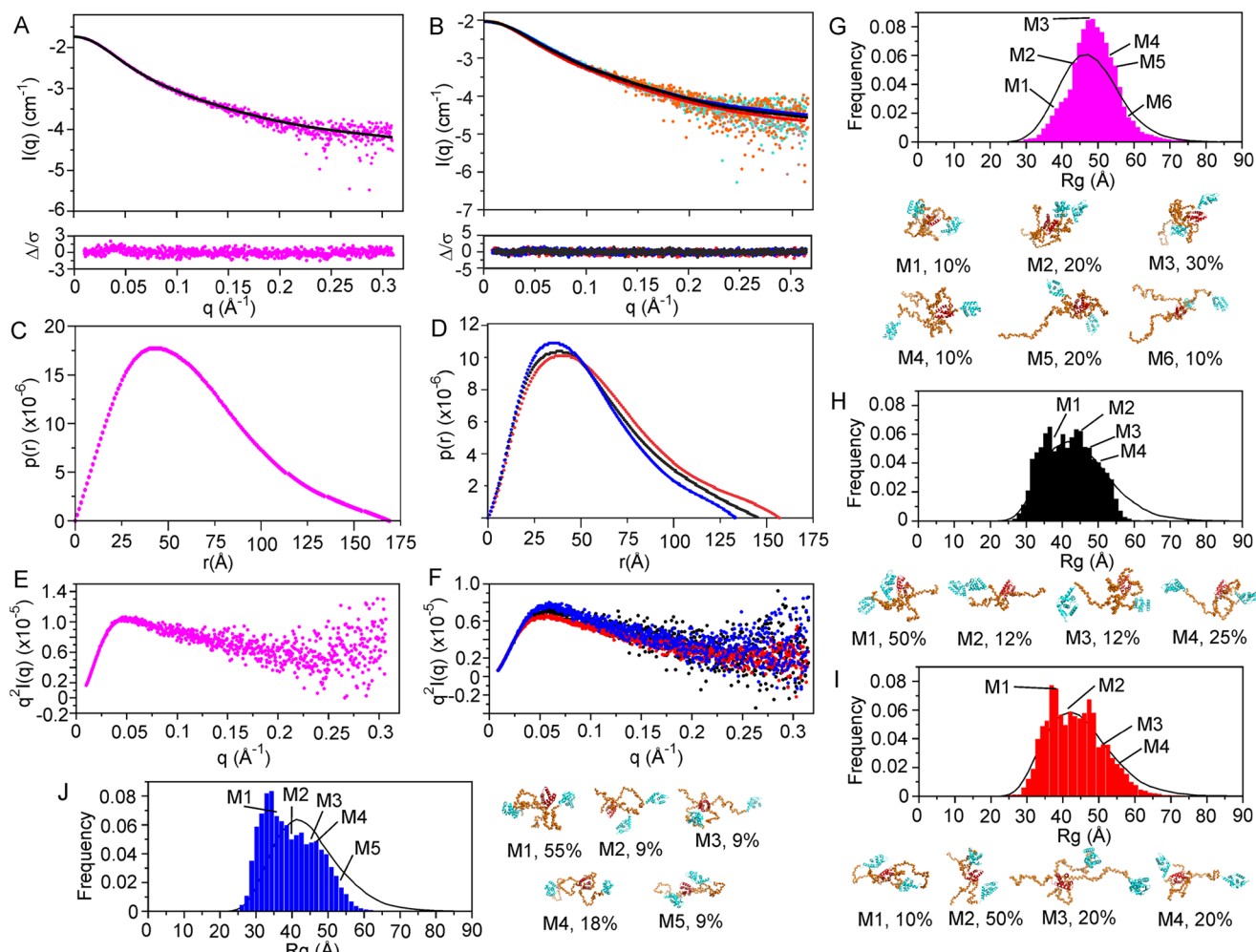

**Fig. 5 | P1 and P3 wt, KRm and D289N show long-range conformational differences.** Small Angle X-Ray Scattering data for (**A**) P1 (black, magenta) and (**B**) wt P3 (black, brown), P3-KRm (red, orange) and P3-D289N (blue, cyan) and the corresponding p(r) curves for (**C**) P1 (magenta) and (**D**) wt P3 (black), P3-KRm (red) and P3-D289N (blue). The p(r) curve for P3-D289N and P3-KRm are consistent with respectively smaller and larger radius of gyration compared to wt P3 (Supplementary Table 1). The Kratky plots for (**E**) P1 (magenta) and (**F**) wt P3 (black), P3-

KRm (red) and P3-D289N (blue) rise to plateaus which are representative of proteins that contain significant regions of disorder. EOM modeling distributions of (**G**) P1 (magenta), (**H**) wt P3 (black), (**I**) P3-KRm (red) and (**J**) P3-D289N (blue) show P1 and the P3 proteins are distributed between open and closed conformers with P3-D289N showing a greater preference for closed and P3-KRm for open relative to wt P3.

IDR1 (Lys54) and IDR2 (Lys155, Lys170), two between IDR1 (Lys54, Lys76) and the CTD (Lys231, Lys285 respectively), two between IDR2 (Lys144, Lys170) and the CTD (Lys242, Lys285 respectively), one between the DD (Lys116) and IDR1 (Lys54) and one within IDR1 (Lys54 to Lys62). The numerous additional long-range cross-links in P1 originating from the IDR1, that are absent in P3 support closed conformations that predominate in P1 compared to P3. Notably, a cross-link between Lys214 to Lys282 of the CTD in P3, which are 15 Å apart in this structured domain, is consistent with an intra-protomer interaction. However, it is absent in P1 which suggests that it may also be inter-protomer due to conformational flexibility in P3.

The comparatively more compact structure of P1 as suggested by EOM and XL-MS, highlights a higher degree of conformational diversity of the N-terminally truncated isoform P3 compared to the full-length P1, which could potentially contribute to the diversity of functions demonstrated for P3. To model possible structural differences between P1 and P3, we generated AlphaFold models. The predicted P1 and P3 dimers show the expected multi-domain organization for each protomer (Supplementary Fig. 10A, B); however, the pTM scores for the generated models of P1 and P3 were ~0.3, which indicates a low confidence. The TM alignment scores for the ordered DD and CTD of

the predicted structures with the experimental structures of CVS DD (**8FUQ**) and CVS CTD (**1VYI**) were greater than 0.9, which indicates good structural alignment between predicted and experimental structures and that the overall topology of these domains are preserved. Hence, the low pTM scores for the AlphaFold models of P1 and P3 are due to the presence of IDR1 and IDR2 (fraction of disorder in the models is ~0.36 for P3 vs ~0.4 for P1) (Supplementary Figure 10A, B). This is also observed in the pLDDT confidence scores, where the DD and CTD of predicted P1 and P3 structures are high (>90), but the IDR1 and IDR2 have low pLDDT scores (<50). These models propose for P1 that IDR1 interacts extensively with the DD and CTD. The CTD and DD of the P3 model maintain an interaction with the IDR1; however, as IDR1 is truncated, this is consistent with a weaker interaction. An important difference in the models is that for P1 the CTD is oriented whereby the positively charged face (where Lys214 and Arg260 are located) points inwards to the overall structure restricting access to these residues (Supplementary Fig. 10A) whereas in the P3 model, this positively charged face of the CTD points outwards and is exposed (Supplementary Fig. 10B). Nevertheless, it is important to note that these static models of P1 and P3 do not reflect the conformational heterogeneity observed in the SAXS data. Additionally, most predicted cross-links of

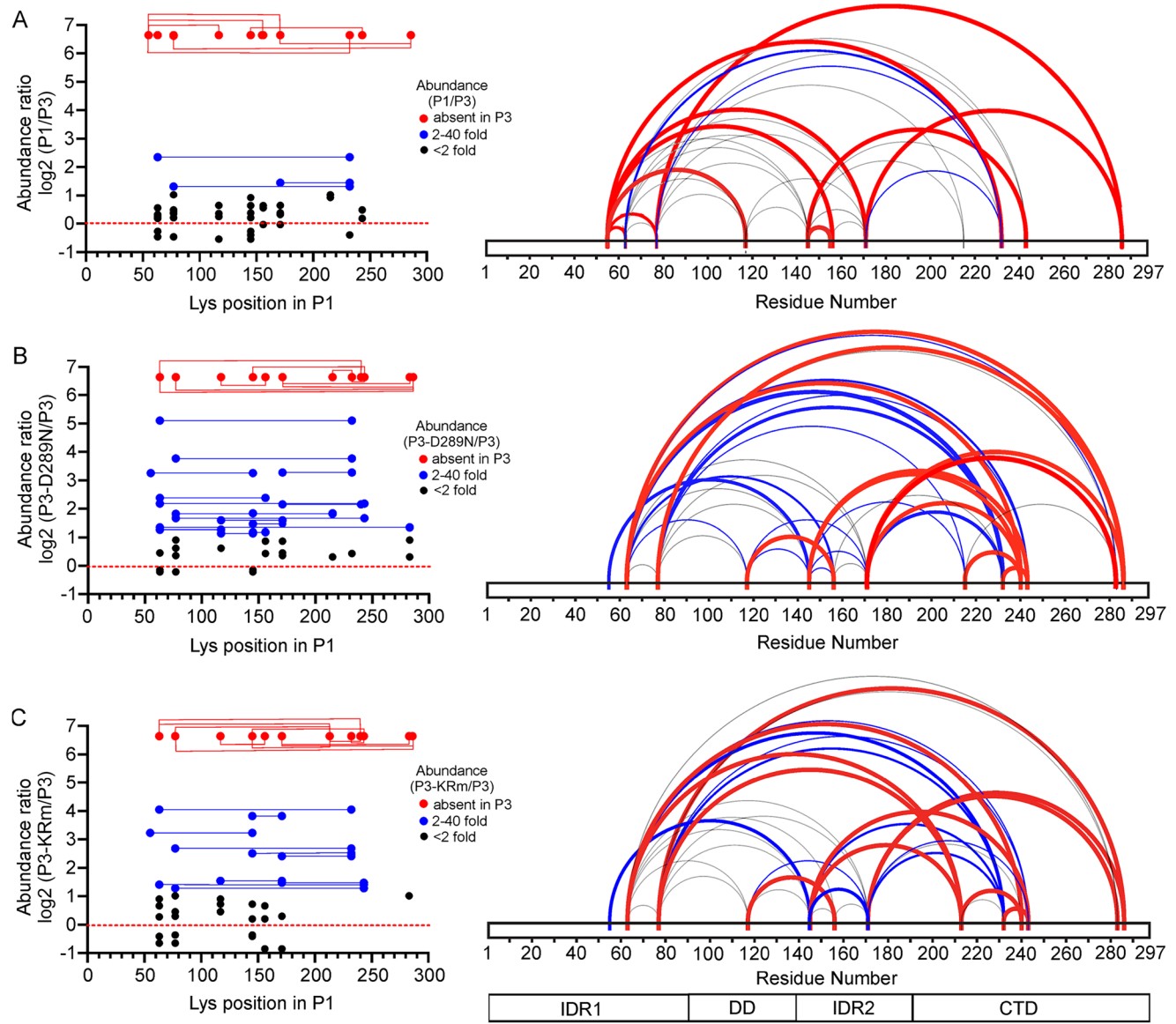

**Fig. 6 | CVS P3-KRm and P3-D289N compared to wt P3 show long-range conformational differences.** Plots of log2 of relative abundance of cross-link patterns for (**A**) P1, (**B**) P3-D289N and (**C**) P3-KRm relative to wt-P3. On the left are plots of abundance ratios, where red represents cross-links absent in P3 but clearly abundant in the protein being compared. In gray are cross-links less than two-fold different in abundance; blue cross-links greater than two-fold more abundant but less than 40-fold. On the right are the cross-links coded by color and line-thickness to correspond to the abundance ratios. Below is a strip diagram of the domain structure of P protein (IDR1, IDR2, DD and CTD are indicated).

the XL-MS analysis are not observed in these models, consistent with the low confidence of the modeling of IDR1 and IDR2.

**Structure of P3 loss- and gain-of function mutants**

To determine the effects of the mutations of P3-KRm and P3-D289N on P3 structure, we used SAXS (Fig. 5, Supplementary Fig. 9, Supplementary Table 1) and XL-MS (Fig. 6B, C). Prior to this analysis, as these mutations are within the structured CTD, we used chemical shift differences and hydrogen-deuterium exchange by NMR to assess the structural integrity of the isolated CTD, finding that the mutations had little impact on stability and fold of the CTD (Supplementary Fig. 11 and Supplementary Table 2). Thus, the loss or gain of function observed in P3-KRm and P3-D289N, respectively (e.g., Figures 1, 3), does not appear to be due to a local impact on domain structure, but rather is likely due to the broader intra- or interprotomer interactions of the domains and disordered regions of P3. XL-MS data for both P3-KRm and P3-D289N (Fig. 6B, C) compared to wt P3 showed significant differences in the

number of cross-links, suggesting that the mutants differ in their overall conformations to wt P3. Notably, Lys212 of P3-KRm and Lys214 of P3-D289N both show cross-links to Lys239, which are absent in wt P3. Upon structural inspection of the CTD, it is evident that Lys212 and Lys214 are on the opposite face of the CTD and greater than 27 Å from Lys239, suggesting that these cross-links are inter-and not intra-protomer; therefore, we cannot rule out that other cross-links are due to conformational heterogeneity that favors inter-protomer interactions.

There are additional distinct crosslink patterns for each mutant distinguishing their conformational ensemble from that of wt P3. P3-D289N shows cross-links from Lys285 of the CTD to Lys62 and Lys76 of IDR1 which are not observed in wt P3 (Fig. 6B). The cross-link Lys285 to Lys76 is also observed in P3-KRm, but not the cross-link between Lys285 to Lys62 (Fig. 6C). In P3-KRm, there are cross-links between Lys212 of the CTD to Lys62 and Lys76 of IDR1, and Lys144 of IDR2, which are not observed in wt P3 or P3-D289N (Fig. 6B, C). Indeed, no significant crosslinks are observed to Lys212 in wt P3, P3-D289N or P1.

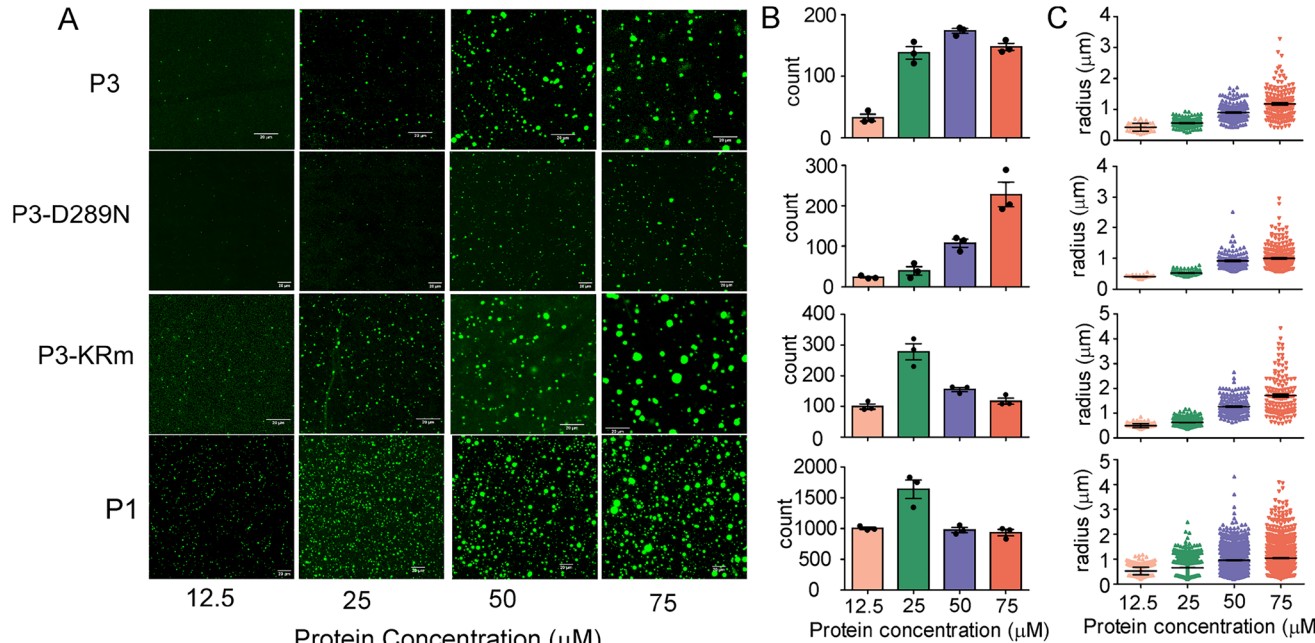

**Fig. 7 | P1, wt P3 and P3 mutants show in vitro phase separation. A** CLSM images show that P1, wt P3, P3-D289N and P3-KRm with 10% PEG 6000, form phase-separated droplets. P1 and P3-KRm, compared to wt P3, show enhancement of phase separation (higher number of droplets (count) and larger radius of droplets at equivalent protein concentrations), whereas P3-D289N shows a reduction in phase separation at the protein concentrations tested (12.5 to 75 μM). Note that droplet count and size (apparent radii of droplets) distribution is dependent on protein concentration with a tendency to form larger coalesced droplets with a net lower droplet count at high concentrations, indicative of droplet fusion especially for P1 and P3-KRm. Images are representative of three fields of view, scale bar: 20 μm. **B, C** Images such as those in (**A**), representative from 3 fields of view, were used to estimate (**B**) droplet count (mean ± SEM, *n* = 3 fields of view) and (**C**) size distribution (radius) with mean ± SEM indicated for droplets determined using the in-built particle analyzer algorithm in ImageJ (Fiji) (see "Methods"). Source data are provided as a Source Data file.

These cross-links may be enhanced in P3-KRm due to the mutation at Lys214 affecting how the CTD orients to IDR1 and IDR2. Notably, a 1.9- and 3.5-fold more abundant cross-link is observed between Lys214 and Lys144 in P1 and P3-D289N respectively compared to wt P3 (Supplementary Data 2). In P1, Lys54 of IDR1 shows numerous cross-links to lysines in IDR2 and the CTD that are mostly absent in wt P3 and the two mutants of P3, supporting more compact/closed structures for P1 (Fig. 6A). The crosslink of Lys54 to Lys144 of IDR2 is present in wt P3, but is > 20-fold weaker than in P1 (Supplementary Data 2). This cross-link is present in both P3-KRm and P3-D289N where it is 9-fold more abundant than in wt P3. In addition, for P3-D289N, Lys62 of IDR1 cross-links to Lys155 of IDR2 where it is ~5-fold more abundant in P3-D289N, and only ~1.5-fold more abundant in P1 and P3-KRm, compared to wt P3. Together, these data suggest that the conformational diversity for all four proteins differs, and that both the KRm and D289N mutations within the CTD have affected interactions between the NTR and the CTD either within or between protomers.

We attempted modeling of the two mutants using Alphafold; however, as expected, they were indistinguishable from wt P3. Therefore, to provide further supporting data for the conformational differences of the two mutants, we acquired SAXS data of the two mutants. P3-KRm (Fig. 5I) has a more elongated envelope shape compared to wt P3, indicated by the higher $R_g$ of 46.12 Å and $D_{max}$ of 158.86 Å of P3-KRm, compared to the $R_g$ of 41.60 Å and $D_{max}$ of 144.38 Å of wt P3 (Figs. 5D, 5F, Supplementary Fig. 9; Supplementary Table 1). In contrast, SAXS analysis of P3-D289N (Fig. 5J) suggested slightly more compact/closed structures supported by the smaller $R_g$ of 40.01 Å and $D_{max}$ of 133.22 Å compared to wt P3 (Fig. 5D, F; Supplementary Table 1). EOM analysis also supports that P3-KRm shows a preference for more open conformations, and P3-D289N for more closed conformations, compared to wt P3 (Fig. 5H–J). Considering the major impact of these mutations on the characteristic phenotypes of

P3, these findings emphasize roles for interaction of the NTR and CTD in the unique functions of P3, supporting the hypothesis that conformational organization of the domains underpins multifunctionality and the unique functions of different isoforms.

### P1 and P3 form liquid bodies in vitro

P protein contains two IDRs (Fig. 1A), which, when P is in complex with N-protein in cells, are essential for LLPS to form the viral Negri body MLOs[19]. Recently, P1 has been shown to independently phase separate in vitro[59]. P3 also contains two IDRs and, as shown here, additionally interacts with a diverse subset of cellular MLOs independently of other viral proteins[12,23,24]. Recently, Nishigahara P3 was shown to form liquid droplets in vitro (visualized by light microscopy)[26]. Droplet formation was impaired in Ni-CE P3 by substitutions affecting N- and C-terminal region interactions and P3 functions, indicating that, dependent on conformational organization, P3 undergoes LLPS enabling partitioning into cellular MLOs[26]. To define the LLPS properties of P3 and compare the LLPS characteristics of P1 and wt and mutant P3, we used confocal microscopy to analyse the P proteins tagged with the small fluorescent label, BODIPY R6G. Wt P3 readily forms liquid droplets under physiological conditions in vitro (Fig. 7), albeit less efficiently than P1 in terms of both droplet number and radius, but consistent with an intrinsic capacity for LLPS[26,59,60] and capability to form interactions with multiple MLOs in cells (Fig. 3). Thus, it appears that, while intrinsic LLPS is likely involved in MLO interaction by P3, the capacity to undergo LLPS does not account for differing tendencies of P1 and P3 to associate with cellular MLOs; in fact, the capacity for LLPS and MLO association is opposite to that which might be expected. Thus, these differences are likely to relate to differing interaction of P1 and P3 with other component(s) of cellular MLOs.

To examine whether the structural differences induced by the mutations KRm or D289N, and associated loss or gain in MLO

interactions relate to altered intrinsic LLPS capacity, we assessed their droplet formation. Both P3-KRm and P3-D289N readily formed droplets under physiological conditions (Fig. 7), with P3-KRm showing a marked increase in LLPS capacity compared with wt P3, in both droplet count and radius. In contrast, P3-D289N showed a reduction in droplet radius, although, at higher protein concentrations, droplet count was similar to wt P3. Thus, similar to P1, the loss of MLO interaction by P3-KRm, and gain of interaction by P3-D289N, does not correlate with major changes in intrinsic LLPS propensity and appears opposite to that expected. As IDRs are typically associated with LLPS, the data for P3-KRm would appear to be consistent with the retention of interactions between IDR1 and 2 in this mutant[26].

## P3 binding to RNA correlates with unique phenotypes

Our data indicate that P1 and P3 interact similarly with well-defined MLO-associated cellular proteins following solubilization of cells (Fig. 4), and that both proteins undergo LLPS in vitro (Fig. 7), with P1 LLPS being notably more efficient, indicating that these properties do not directly account for differences in MLO associations. The fact that the P3 mutations have general impacts on multiple different P3-MLO associations (Fig. 3), which do not correlate with major effects on LLPS propensity (Fig. 7), suggests that interactions with other MLO components may be important. We thus hypothesized that interactions with RNA, which often contribute to protein recruitment into cellular MLOs (e.g., nucleoli, NBs, MT bundles[61]), might be a key factor and could account for the generalized effects of mutations on interactions of P3 with different MLOs[50]. Supporting this, we observed that RNAse A treatment of living cells expressing GFP-P3 led to a rapid loss of P3 from nucleoli (Supplementary Fig. 12A). Furthermore, treatment with Actinomycin D (ActD; which inhibits rRNA transcription and induces nucleolar stress[62]) resulted in relocalization of P3 to nucleolar caps (Supplementary Fig. 12B), a characteristic outcome of disrupted RNA-mediated nucleolar interactions[63]. Together, these findings suggest a role for RNA in mediating P3 association with MLOs.

P1 associates with LLPS-dependent Negri bodies, which contain viral and cellular proteins, RNA genome and transcripts; however, Negri body association of P1 in cells depends on co-expression of N protein indicating that N protein interaction is required for P1 LLPS and MLO association in cells[19,59]. P1 liquid droplets formed in vitro have been reported to associate with poly-A RNA based on accumulation of fluorescence (using a fluorescent dye to label RNA) detected by microscopy[59]; however, such interactions have not been assessed using non-labeled RNA, and no comparative analysis of P1 and P3 for RNA interaction has been reported. The capacity of P3 to interact independently of other viral proteins with cellular MLOs suggests that it may be adapted (through loss of the N-terminal region and associated effects on N- and C-terminal interactions) to bind to RNA and enter MLOs, with potential importance to viral replication (e.g., through potential regulatory interactions of P3 with Negri bodies) and/or host subversion. The latter would provide a mechanism for the formation of the broad virus-host intracellular interface.

To assess and compare the interaction of P3 and P1 with RNA, we used agarose gel electrophoresis and size exclusion chromatography (SEC) to initially examine RNA binding by wt P3, P3 mutants and P1, following expression and purification from *Escherichia coli* in the presence or absence of RNase A. Wt P3 co-purified with nucleic acid that was lost following RNase A treatment (Supplementary Fig. 13A), but no nucleic acid co-purified with P1. Thus, despite similar interactions with MLO-associated proteins in IPs from cell lysates, P1 and P3 appear to differ substantially in interaction with RNA. P3-KRm also lacked nucleic acid interaction, but P3-D289N consistently co-purified with RNA, and greater amounts of RNA were copurified with P3-D289N than with wt P3. SEC analysis indicated that the apparent size of purified proteins in the absence of RNase A differed, consistent with the differential interaction with RNA indicated by gel electrophoreses (i.e.,

P3-D289N was largest, P3-KRm smallest, wt P3 intermediate) (Supplementary Fig. 13B). Following RNase treatment, all proteins eluted at the same volume supporting that the differing sizes reflect differential interaction with RNA (Supplementary Fig. 13C).

To further assess interactions by these P proteins with RNA under LLPS conditions we used a turbidity and centrifugation-based saturation concentration (Csat) assay[64] to separate the light and dense phases, in the presence and absence of RNA (20 nucleotide sequence of polyA as previously used in a fluorescently-labeled form to indicate interaction with P1 droplets generated in vitro in a crowded environment[59], and torula yeast tRNA, representing a more complex RNA mixture). Firstly, we constructed phase diagrams for P1, wt P3, P3-D289N and P3-KRm, in the absence of RNA (Fig. 8A). Consistent with the confocal microscopy (Fig. 7), P1 showed a trend to phase separate more readily than the P3 constructs, emphasizing that differing intrinsic LLPS capacity per se cannot account directly for differing MLO association profiles. Working at 100 µM of protein and 10% PEG 6000, we next conducted turbidity assays in the presence of RNA. Upon the addition of equimolar torula yeast tRNA, there was no significant increase in turbidity (Fig. 8B). However, on separation of the light and dense phases by centrifugation, and using agarose gel electrophoresis to detect RNA, torula yeast tRNA was clearly observed in the dense phases of P3-D289N and, to a slightly lesser extent, wt P3 (Fig. 8C). Torula yeast tRNA was weakly observed in the dense phase of P3-KRm and very weakly in that of P1, suggesting little or no association. Notably, in contrast to previous analysis using fluorescently labeled poly-A RNA to assess association with P1 liquid droplets in vitro[59], polyA was not observed to associate with the dense phase of any of the P proteins.

To more accurately assess association of torula yeast tRNA and the P proteins, we modified the Csat assay to monitor the $A_{260}/A_{280}$ ratios of the light phase after centrifugation (Fig. 8D; and Supplementary Fig. 14). In this assay, in the presence of appropriate amounts of torula yeast tRNA and under LLPS conditions, c. 60% of the protein is in the dense phase after centrifugation (estimated by Csat assay for 100 µM protein in the absence of RNA). Hence, $A_{260}/A_{280}$ readings of the light phase will show whether RNA is pulled down by protein or not, that is an $A_{260}/A_{280}$ value of the light phase much greater than that obtained under non-LLPS conditions for the same solution indicates an inability of the protein to sequester RNA into the dense phase.

Firstly, under non-LLPS conditions we titrated 100 µM protein with torula yeast tRNA, to attain a stable $A_{260}/A_{280}$ reading of ~0.6, finding that ratios of 1:0.25 or lower of protein:torula yeast tRNA met this condition (Fig. 8D). Upon addition of 10% PEG, which induced LLPS, and in the presence of yeast torula tRNA, the light phase of P3-D289N and wt-P3 maintained an $A_{260}/A_{280}$ of ~0.6, whereas the $A_{260}/A_{280}$ of the light phase of P3-KRm and P1 increased to ~0.9. These data suggest that for wt-P3 and P3-D289N, yeast torula tRNA was pulled down along with protein in a proportional amount in the dense phase, keeping the $A_{260}/A_{280}$ ratio of the light phase consistent before and after phase separation, whereas very little yeast torula tRNA was pelleted in the dense phase of P3-KRm and P1, leading to an enrichment of the RNA in the light phase. This supports the agarose gel results that P3-KRm and P1 do not sequester RNA into the dense phase (Fig. 8C). Thus, a key distinguishing feature of P3 compared to P1 is the capacity of P3 to bind to RNA, indicating that a major effect of N-terminal truncation is gain of RNA interaction. The finding that C-terminal mutations also regulate RNA interaction is consistent with the significance of the interaction of C- and N-terminal regions of the P proteins (Figs. 5, 6) to modulate this function. Together, the data support a mechanism whereby viral protein-RNA interaction, dependent on conformational organization of the protein isoform, is a critical component and regulator of the formation of the virus-host interface by P protein.

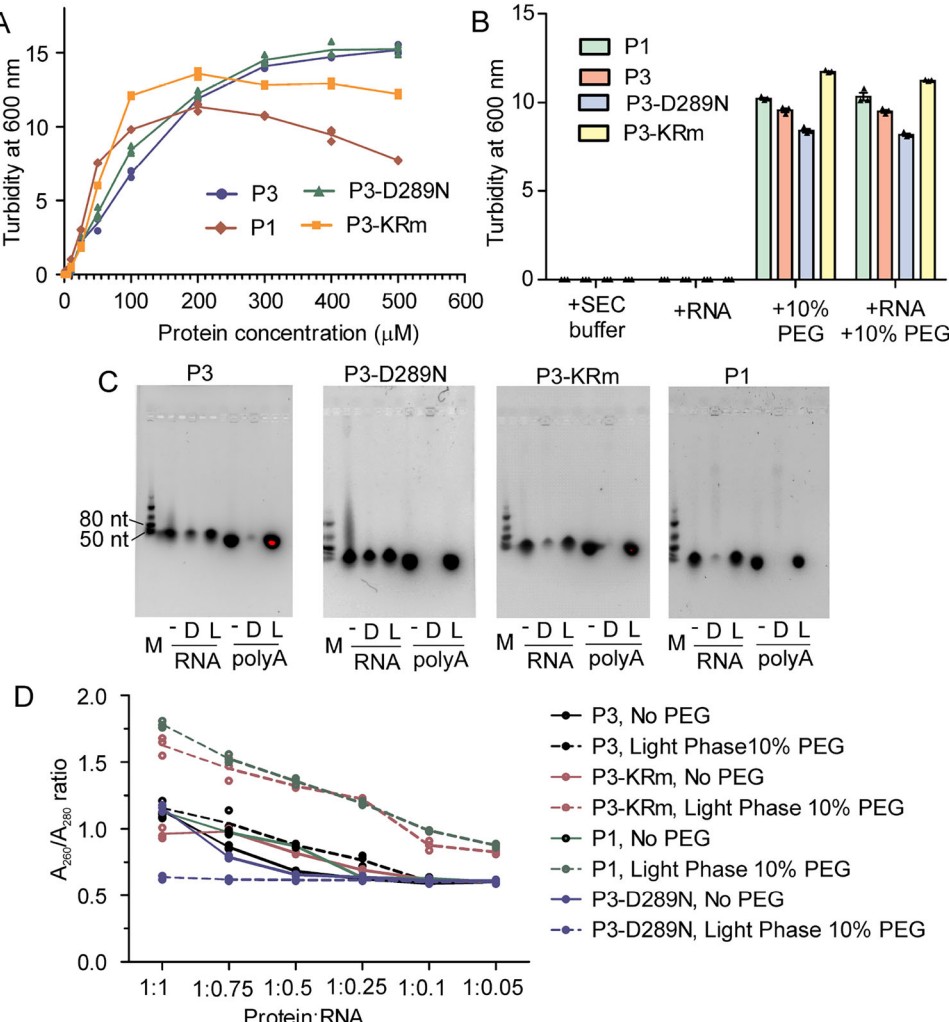

**Fig. 8 | wt P3 and P3-D289N bind RNA but P1 and P3-KRm do not. A** Phase diagrams of wt P1, wt P3, P3-KRm and P3-D289N, constructed by measuring turbidity at 600 nm at 1-500 μM concentration of proteins in the presence of 10% PEG 6000. All proteins show a concentration-dependent increase in turbidity. **B** The different proteins at 100 μM do not show detectable turbidity in buffer (+ SEC buffer) or in the presence of torula yeast tRNA (+ RNA). On the addition of 10% PEG 6000, turbidity for all proteins increased and was not significantly altered by the addition of tRNA (mean turbidity ± SEM, $n = 3$; comparison of samples with and without tRNA used a 2-way ANOVA with Bonferroni's multiple comparison test; no significant difference ($p > 0.05$) was observed). **C** Images of agarose gel analysis of the dense and light fractions of the indicated proteins following incubation with tRNA (RNA) and polyA (see "Methods"). M, marker; -, protein/RNA not phase separated (no PEG); $D$ dense phase after phase separation (10% PEG); $L$ light phase after phase separation (10% PEG). **D** $A_{260}/A_{280}$ ratio of the light phase measured after centrifugation in the absence of PEG (No PEG, solid lines) or with 10% PEG (dashed lines), at decreasing concentration of torula yeast tRNA with a fixed concentration of protein (molar ratios of protein:RNA from 1:1 to 1:0.05). For P3, the $A_{260}/A_{280}$ ratio reaches 0.6 at the protein:RNA ratio of 1:0.1, whereas for P1 and P3-KRm the $A_{260}/A_{280}$ ratio remains greater than 0.6 at all ratios, indicating RNA sequestration into the dense phase by P3, but not by P1 and P3-KRm. For P3-D289N, the $A_{260}/A_{280}$ ratio of the light phase is 0.6 at all protein:RNA ratios, indicating a higher propensity to sequester RNA than wt P3. Data shows mean $A_{260}/A_{280}$ ratio ± SEM ($n = 3$). Source data are provided as a Source Data file.

To further assess this mechanism, we modeled P3 bound to torula yeast tRNA using Alphafold (Supplementary Fig. 10C). The resulting model shows RNA binds between the CTDs of the two protomers where Lys214 and Arg260 directly contact the RNA, consistent with the mutations of Lys214 and Arg260 of P3-KRm directly impacting on the ability of P3 to bind RNA. Notably, this interaction depends on the unique orientation of the CTDs within the P3 dimer whereby the charged residues are exposed in P3, but oriented inward restricting access in P1 such that altered conformation of the CTDs relative to one another is important; thus, the interactions of the NTR and CTD are likely to regulate RNA binding by potentially occluding the partial RNA-binding sites in the individual CTDs, and producing a P1 closed conformation where a functional RNA binding site that is formed through an exposed orientation of two CTDs in P3 is not generated. The model also suggests that the basic-rich region of the DD may also play a role in

RNA binding, although this remains to be tested. Asp289 of the CTD in the model of P3 bound to RNA is also oriented towards the RNA, whereby removal of the negative charge in P3-D289N may enhance RNA binding.

## Discussion
The limited coding capacity of RNA viruses poses significant evolutionary challenges for the generation of proteins sufficient to mediate replication and assembly, and modulation of multiple host cell processes. Here, we have characterized mechanisms by which different protein isoforms expressed from the single P gene of RABV achieve multifunctionality, including insights into how apparent increases in functionality arise in truncated isoforms, despite their having identical sequences other than the loss of N-terminal regions, which would be expected to remove rather than add functions. Specifically, higher-

order organization involving extensive non-sequential contacts between sequentially distant regions (including well-separated IDRs and structured domains) enables interaction of the N- and C-terminal regions that results in positive or negative regulation, generating distinct 'species' with highly divergent phenotypes. This three-dimensional picture supports a mechanism whereby conformational diversity in protein isoforms, P1 and P3, can augment well-defined mechanisms of viral gene multifunctionality[65], such as isoform expression by RNA editing, alternative start sites and overlapping/overprinted sequences, to significantly increase the effective functional coding capacity of viral genomes. Importantly, this conformational diversity provides a molecular basis for observations that unique phenotypic properties and interactions of P3, which have been mapped to the CTD or extended N-terminal sequences, are regulated by the NTR, such that NTR truncation results in a gain rather than a loss of function in P3 compared with the longer P1 protein[21,23,25]. Notably, RNA binding, rather than classical protein-protein interactions, appears to be central to the altered phenotype, as it is evident in P3 but not P1 and correlates with unique P3 functions, where mutations in the CTD that impact on RNA-binding similarly affect the ability of P3 to interact with cellular LLPS structures and mediate associated functions, including MT-dependent immune evasion[11,21]. This provides insights into how P protein isoforms, and probably other viral proteins, can form interfaces with host factors to subvert host cell biology, involving not only conventional protein-protein interactions, but also protein-RNA and, consequently, MLO interactions, regulated by protein conformation.

Our findings show that P3 incorporates features typical of MLO-associating proteins, including high prevalence of IDRs for LLPS, and RNA-binding. P1, P3 and mutants thereof all retain an intrinsic capacity to undergo LLPS in vitro, presumably involving interactions of IDR1, IDR2 and the CTD, such that the capacity for LLPS does not correlate with RNA binding or with differing MLO association; this is emphasized by the fact that P1 and P3-KRm, which lack MLO/RNA interaction, appear modestly enhanced for LLPS in vitro. However, LLPS capacity per se appears to be required for MLO association, as would be expected, indicated by our prior study showing that impaired cellular MLO association by the Ni-CE P3 protein correlates with loss of in vitro LLPS[26]. Furthermore, P3 and P1 form similar interactions with MLO-resident cellular proteins (Fig. 4, and Supplementary Data 1), indicating the common presence of domains or sequences to interact, directly or indirectly, with these proteins. It appears, however, that the interactions of P1 occur post-lysis (disrupting cells and subcellular organelles/MLOs), as P3 alone localizes to the relevant structures in intact cells, and colocalizes with interactor proteins therein. Clearly, the protein interactions we assessed, and intrinsic capacity to undergo LLPS, are insufficient for separation into cellular MLO environments in the absence of interactions with other biomolecules, particularly RNA, which appears to be essential and to provide a key regulatory mechanism between isoforms.

RNA binding is characteristic of many MLO-associating proteins, as weak multipartite protein-protein and protein-RNA interactions are important to MLO formation[66,67]. The finding that P3 mutants affecting RNA interaction also have a general impact on the association of P3 with several distinct MLOs suggests that RNA interaction is a common requirement, so that the P3-RNA interface enables broad MLO targeting. However, the MLO interaction profile of P3 is selective, as indicated by a strong, consistent localization within nucleoli, frequent proximal localization to Gem/Cajal bodies, variable association with PML-NBs, and lack of interaction with paraspeckles (Fig. 2). This differing association profile is consistent with LLPS organization of the cell, where, despite the fact that similar biophysical mechanisms (e.g., low affinity protein-protein/RNA interactions) underlie MLO formation and association, specific physical and molecular properties of MLOs differ, indicated by the observation that some MLOs can fuse, while others are immiscible[68]. Furthermore, integration of proteins into

MLOs typically depends not only on protein LLPS, but also on interactions with resident scaffolds proteins[69], so that some proteins are highly selective and others more promiscuous; P3 appears to represent the latter, but is selective, presumably due to specific molecular interactions with MLO components and/or biophysical properties. MLO interactions can also be dynamically regulated by post-translational modifications[70–72], which appears likely for P3 where the flexibility of the IDRs potentially enables switching between interacting states (e.g., the closed and open states). Phosphorylation of the P3 CTD is implicated in P3-MT bundling[12,15,31,36], and our study indicates effects on nucleolar localization (Fig. 1) consistent with such a mechanism.

Although the physiological roles of some MLO interactions of P protein and P isoforms are not currently defined, MLO-resident interactors (e.g., NCL[23], PML[24] and treacle[13]) are implicated in infection, and MT bundling[11,21,26] and PML-NB reorganisation[24] are evident in infected cells and in cells expressing P3 alone. Furthermore, MT-bundling in infected cells and cells expressing P3 alone is similarly impacted by amino acid substitutions present in Ni-CE virus, correlating with altered immune evasion, viral resistance to IFN, and pathogenesis[11,21]. We now also show that protein detected by anti-P-protein antibody can form NBs in infected cells (Supplementary Fig. 4), which is likely to primarily represent P3 as this is the abundant isoform that localizes within the nuclear compartment, supporting roles for such interactions during infection. Thus, it appears that LLPS and/or RNA interactions are major drivers in the formation of important elements of the virus-host intracellular interface.

Our findings relating to LLPS in the formation of viral protein-host MLO interfaces prompt the question of how such interfaces developed. Several proteins of RNA viruses, including RABV P protein, use LLPS to form replication factories, which are analogous to cellular MLOs in separating and organizing processes[19,73–76], including orchestrating genome replication/assembly and segregation from the immune surveillance machinery. RABV was the first of several RNA viruses shown to replicate in liquid organelles[19], specifically Negri bodies, the formation of which requires the full-length P1 (and P1 IDRs) and N protein[77]. MLOs have been proposed to be descended from pre-biotic assemblies of the "RNA world"[78,79] and similar concepts are suggested for some RNA viruses[80]. Thus, cells may be mimicking viral mechanisms of molecular organization by LLPS rather than vice versa. It seems likely that RNA viruses have evolved proteins with intrinsic LLPS propensity to form replication factories, and that accessory proteins such as P3, which are not essential for replication but share sequences/structures with replication proteins, have inherited these properties, and evolved additional properties (e.g., N-protein-independent RNA interaction by P3) to enable cellular MLO targeting to form large interaction arrays[21,23,24]. The extension of viral-protein LLPS beyond replication factories would provide mechanisms whereby the limited complement of relatively small accessory proteins typical of RNA viruses could modulate multiple processes including immunity, stress responses and metabolism in host cells. The absence of enzymatic or structural roles for P gene products in the virion likely allows greater evolutionary flexibility, enabling the expression of multiple isoforms, diversification of sequence and structure, and the emergence of conformationally distinct forms that support multifunctionality.

## Methods

### Cells, plasmids and transfections
HeLa (ATCC CCL-2) and HEK-293T (ATCC CRL-3216) cells were maintained in Dulbecco's Modified Eagle Medium (DMEM, ThermoFisher Scientific, Cat# 11965092) supplemented with 10% Fetal Calf Serum (FCS) at 37 °C, 5% $CO_2$. All CVS P protein mammalian expression plasmids were cloned into pEGFP-C1 were generated previously[12,20,31], except P3 mutants K282N, K285N and D289N, which were generated in

this study by overlap extension PCR using the primers 5′-gaggctga-caatctaagcaaaatcatgcaagat-3′ (K282N), 3′-gctaagcaatatcatgcaagatgatttgaatc-5′ (K285N) and 5′-catgcaaaatgatttgaatcgctatacatc-3′ (D289N) and sequence verified. Plasmids were transfected into HEK-293T cells and HeLa cells using Lipofectamine 2000™ or Lipofectamine 3000™ respectively, according to the manufacturer's instructions (Thermo-Fisher Scientific). All newly generated plasmids are available from the corresponding author upon request.

## Antibodies and reagents

Antibodies used in this study include NCL (CST, Cat# 14574; IB 1:1000), NPM1 (ThermoFisher, Cat# 32-5200; IB 1:2000), NOLC1 (ProteinTech, Cat# 11815-1-AP; IB 1:2000; Abcam, Cat# Ab204318, IF 1:50), RBM14 (Bethyl Laboratories, Cat# A300-331A; IB 1:2000, IF 1:50), SFPQ (Abcam, Cat# Ab11825; IB 1:1000), NONO (Abcam, Cat# Ab133574; IB 1:2000, IF 1:250), G3BP1 (Santa Cruz, Cat# sc-81940; IB 1:200, IF 1:100), Treacle (ProteinTech, Cat# 11003-1-AP; IB 1:2000, IF 1:100), Coilin (Santa Cruz, Cat# sc-55594; IF 1:200), SMN (BD Biosciences, Cat#610646, IF 1:1000), PML (Santa Cruz, Cat# sc-966; IF 1:50), GFP (Abcam, Cat# Ab6556; IB 1:5000), β-tubulin (Sigma, Cat# T8328; IB 1:5000; IF 1:1000), UBF1(Abcam, Cat# Ab244287; IF 1:500), Rabies P-protein (Cat#MBS1489481; IF 1:200 for untagged P-protein staining). Secondary antibodies (1:10,000 dilution) used for IB were goat anti-rabbit (Merck, Cat#AP307P) and goat anti-mouse (Merck, Cat# AP308P) IgG horse radish peroxidase (HRP)-conjugated-antibodies. Secondary antibodies (1:1000 dilution) used for IF were goat anti-rabbit Alexa Fluor 568 (Cat #A-11011) and goat anti-mouse Alexa Fluor 568 (Cat #A-11004), purchased from Thermo Fisher Scientific. To inhibit CRM1-mediated nuclear export, cells were treated with leptomycin-B (LMB, Cell Signaling, Cat#9676; 2.8 ng/mL) for 3 h prior to analysis.

## Confocal laser scanning microscopy (CLSM) and image analysis

CLSM used a Leica SP5 or Nikon C1 microscope with 60× oil immersion objective (NA 1.4) and a heated chamber (37 °C) for live-cell analysis. Photobleaching of cytosolic GFP was performed using Nikon C1 microscope. A region of interest was selected in the cytoplasm and repeated bleaching (100%, 488 laser, 10 s) and recovery (10 s off) was performed over the course of 5-10 min (depending on fluorescence level of the particular cell) until levels of GFP were significantly reduced. For analysis of effects of RNase A on the localization of GFP-fused protein in live-cells, a modified protocol of Zhou et al. [81] was used: HeLa cells transfected to express GFP fused proteins were permeabilized with 1% Tween20 for 10 min before careful removal of the buffer. Cells were briefly washed with PBS, and then treated with 200 µg/ml RNAse A (Qiagen) in PBS with 5 mM MgCl₂ and 100 mg/mL bovine serum albumin (BSA), before imaging by CLSM over a 10 min period.

For IF staining, cells seeded onto glass coverslips were fixed with 4% paraformaldehyde (37 °C, 10 min) unless otherwise indicated, permeabilized with 0.25% Triton X-100 (room temperature (RT), 5 min), and blocked with 1% BSA in PBS (RT, 1 h), before primary and secondary antibody labeling (RT, 90 min each); coverslips were mounted onto glass slides with Mowiol. For ActD-treated cells, cells were treated with 0.01 µg/mL ActD (which, at low levels, shuts down RNA Pol I, but not RNA Pol II)[62] for 4 h prior to fixation with ice-cold methanol (10 min), and IF staining for UBF1.

Image analysis was performed using Fiji (ImageJ) freeware software. To determine the nucleolar to nuclear fluorescence ratio ($Fnu/n$), the mean fluorescence intensity in the nucleolus ($Fnu$), nucleus ($Fn$) (Hoechst 33342 staining of DNA was used to help identify nuclei and nucleoli for live and fixed cells) and background ($Fb$; untransfected cells) was determined and the $Fnu/n$, calculated using the formula: $(Fnu − Fb)/(Fn − Fb)$. To determine the prevalence of P3-MLO associations, CLSM images of living cells transfected to express the

indicated P3 constructs were analysed to determine (1) % of cells with apparent MT association, as assessed by apparent cytoplasmic P3 filaments, (2) % of cells with clear nucleolar association, determined by accumulation of fluorescence within nucleoli (i.e., nucleolar fluorescence ≥ nuclear fluorescence), (3) % cells with P3 NBs, determined as clear punctate fluorescent nuclear structures not associated with nucleoli.

## Rabies-infected cell imaging

HEK-293 cells were infected with rabies virus (CVS-11 strain) at a multiplicity of infection (MOI) of 2. At 48 h post-infection (p.i.), cells were fixed with ice-cold methanol for 10 min, followed by fixation with 4% PFA for at least 1 h to ensure proper decontamination before removal from the BSL-3 containment facility. Fixed cells were immunostained using an anti-P antibody (M2418[82], diluted 1:10; a kind gift from Dr. Christine Fehlner-Gardiner, Center of Expertise for Rabies, Canadian Food Inspection Agency) and an AlexaFluor 488-conjugated secondary antibody. Imaging was performed using a Nikon C1 confocal microscope and Zeiss LSM 800 microscope.

## NB fusion imaging

Wide-field imaging of live cells was performed using a Nikon Ti2 microscope, configured to maintain optimal physiological conditions. The microscope was equipped with a stage and objective warmer (Okolab) for control of CO₂ and humidity. Cells were imaged in 1.5 mL of DMEM, pre-heated and maintained at 37 °C, within a controlled environment of 5% CO₂ and 90% humidity for cell viability. Images were captured using a 100x, 1.49 numerical aperture (NA) oil-immersion objective lens. A Lumencor AURA Light Engine, with five independently controllable LED light sources, provided illumination. The intensity of the light from the AURA engine was further controlled and attenuated using neutral density filters to minimize photobleaching and phototoxicity. A Teledyne Kinetix sCMOS camera with a 6.5 µm pixel size and 29.4 mm field of view was used for capturing of images. Time-lapse data was collected by acquiring volumetric data at a rate of 1 volume every 10 seconds over durations ranging from 1 to 2 h\. Each volume consisted of slices acquired with a step size of 0.2 µm, with each slice exposed for 50 ms.

## *d*STORM imaging and analysis

Transfected COS-7 cells were fixed 18 h p.t. with 2% (v/v) glutaraldehyde in cytoskeletal buffer (CB: 10 mM 2-ethanesulfonice acid, 150 mM NaCl, 5 mM ethylene glycol tetraacetic acid, 5 mM glucose, 5 mM MgCl₂) for 10 min, then permeabilised in 0.3 % (v/v) glutaraldehyde/ 0.25 % (v/v) Triton X-100 in CB for 2 min. 0.1% NaBH₄ in PBS was added to fixed cells for 7 min in order to quench residual unreacted glutaraldehyde and reduce autofluorescence during imaging. Fixed cells were then blocked with 5 % (v/v) BSA in PBS for 30 min. MTs were immunolabelled using anti-β-tubulin primary antibodies (Sigma, 1:100 in 5 % BSA/PBS, 1 h) and Alexa Fluor 647-conjugated secondary F(ab')2 antibody fragment (Invitrogen, 1:200 in 5 % BSA/PBS, 45 min). Labeled cells were fixed with 4 % (w/v) formaldehyde in PBS for 10 min.

For *d*STORM, cells were imaged in a switching buffer containing 100 mM mercaptoethylamine in PBS, adjusted to pH 8.5 using 1 M KOH. *d*STORM image acquisition was performed on a home-built single molecule super-resolution microscope (Olympus IX81, 100×1.49 NA TIRF objective, Andor iXon EM-CCD detector) based on a previously described microscope setup[83]. GFP-positive transfected cells were selected under epifluorescence illumination (488-nm Toptica laser diode). High power red laser excitation (638-nm Oxxius Laser-Boxx, ~3 – 5 kW cm⁻²) was used to induce photoswitching of Alexa Fluor 647 and the resulting single molecule emissions were captured at 50 Hz for 10,000 – 20,000 frames. Imaging parameters were controlled using Micromanager image acquisition software.

Acquired frames were processed in rapid*STORM*[84] using a pixel size of 100 nm and a point spread function full width half max (PSF FWHM) of 360 nm. Each detected PSF (i.e., single molecule emission) was fitted with a 2D Gaussian function to retrieve a 2D coordinate of the emitting single molecule. Combining the coordinates from all localized PSFs onto a 2D spatial map constructs the super-resolved *d*STORM image. MT bundling was the quantified as previously described[21].

## Co-immunoprecipitation (co-IP)

All co-IPs were performed using 6 cm dishes of HEK-293T cells transfected to express the indicated proteins and lysed at 24 h p.t. with lysis buffer (10 mM Tris/Cl pH 7.5; 150 mM NaCl; 0.5 mM EDTA, 0.5% NP-40, 1 × Protease Inhibitor Cocktail (PIC; Sigma-Aldrich Cat #11697498001)) for 30 min at 4 °C. Supernatants were collected by centrifugation at 20,000 × *g* for 15 min at 4 °C and 10% of the cleared lysate was collected for 'input' analysis; the remaining lysate was subjected to IP using 10 μL of GFP-Trap® beads (Chromotek). Beads were washed 3 times with dilution buffer (10 mM Tris/Cl pH 7.5; 150 mM NaCl; 0.5 mM EDTA, 1 × PIC). Samples for IB analysis were resuspended in 2 × SDS-PAGE sample loading buffer. Samples for mass-spectrometry analysis were subjected to a final wash containing no PIC.

## Proteomic mass spectrometry and analysis

IP-MS was performed with four biological replicates each for GFP control, P1, and P3 immunoprecipitations. Proteins bound to beads from IP assays were eluted using trifluoroethanol (TFE), followed by pH adjustment to approximately 8.0 using triethylammonium bicarbonate (TEAB) buffer. Proteins were then digested with trypsin for 16 h at 37 °C. Peptides were subjected to C18 cleanup. C18 resin was first activated by adding 180 μL of Buffer B (80% acetonitrile, 0.1% formic acid), followed by centrifugation at 500 × *g* for 3 min. This step was repeated once. After discarding the flow-through, the resin was equilibrated with 180 μL of Buffer A (2% acetonitrile, 0.1% TFA), followed by centrifugation at 500 × *g* for 3 min, also repeated twice. Peptide digests were then loaded onto the equilibrated C18 resin and centrifuged at 500 × *g* for 3 min. The resin was washed twice with 180 μL of Buffer A, followed by centrifugation as before. Peptides were eluted by adding 180 μL of Buffer B and centrifuging at 500 × *g* for 3 min. The eluates were immediately vacuum-dried.

Prior to liquid chromatography tandem mass spectrometry (LC-MS/MS), peptides were resuspended in Buffer A. LC-MS/MS was performed on an Orbitrap Eclipse Mass Spectrometer (ThermoFisher Scientific). The LC systems were equipped with an Acclaim Pepmap nano-trap column (Dinoex-C18, 100 Å, 75 μm x 2 cm) and an Acclaim Pepmap RSLC analytical column (Dinoex-C18, 100 Å, 75 μm × 50 cm). The tryptic peptides were injected into the enrichment column at an isocratic flow of 5 μL/min of 2 % (*v*/*v*) CH3CN containing 0.05 % (*v*/*v*) TFA for 6 min, applied before the enrichment column was switched in-line with the analytical column. The eluents were 0.1 % (*v*/*v*) FA (solvent A) in water and 100 % (*v*/*v*) CH3CN in 0.1 % (*v*/*v*) FA (solvent B), both supplemented with 5 % DMSO. The gradient was at 300 nL/min from (i) 0–6 min, 3 % B; (ii) 6–7 min, 3–4 % B; (iii) 7–82 min, 4–25 % B; (iv) 82–86 min, 25–40 % B; (v) 86–87 min, 40–80 % B; (vi) 87–90 min, 80–3 % B; (vii) 90–90.1 min, 80–3 % B and equilibrated at 3 % B for 10 min before injecting the next sample.

Raw MS data was processed using MaxQuant (version 2.6.1.0)[85] and searched against the human UniProt/Swiss-Prot database (canonical sequences only, downloaded June 21st 2024) with GFP, RABV-P1 and RABV-P3 sequences (CVS-11 strain) added. The default MaxQuant settings were used, including Trypsin/P as the protease (maximum of 2 missed cleavages), Oxidation (M) and Acteyl (Protein N-term) as variable modifications and Carbamidomethyl (C) as a fixed modification (maximum of 5 modifications per peptide), and 1% FDR at the peptide and protein level. Match between runs was enabled. Protein quantitation used the in-built MaxLFQ algorithm, with normalization disabled to account for expected, biologically relevant differences in the total amount of immunoprecipitated material between control and bait samples.

Downstream statistical analysis used Perseus software (version 2.0.11)[86]. Only proteins identified by at least two unique peptides in at least three replicates of at least one bait were included in the analysis. Contaminants, reverse hits and proteins "only identified by site" were also removed from the dataset. LFQ intensity values were log2 transformed, before imputation of missing values using the minimum determined value. Different bait sample sets were compared (GFP vs GFP-P1, GFP vs GFP-P3, GFP-P1 vs GFP-P3) using two-sided, unpaired t-tests, with significant hits identified if *p* < 0.05 and average log2 fold change > 1. For interactors that were significantly different between P1 and P3, it was also required that interactors be significant in the appropriate comparison to GFP to be deemed "confident". PCA, volcano and box plots were generated using the ggplot2 package (version 3.5.1) in RStudio (version 2024.12.0 + 467). Functional enrichment analysis used g:Profiler (version e111_eg58_p18_f463989d; human database)[87].

High-confidence P3 interactors associated with MLOs were identified by cross-referencing our dataset with published proteomic data or lists for nucleoli[88], SGs[89], paraspeckles[90], nuclear speckles[91] and P-bodies[92].

## SDS-PAGE and immunoblotting

Samples were separated on 8, 10, or 12% denaturing gels by SDS-PAGE before transfer to a nitrocellulose membrane using a BioRad Trans-Blot semi-dry apparatus. After blocking (5% non-fat milk in PBS with 1% Tween20 (PBST)), the membranes were incubated with primary antibodies followed by HRP-conjugated goat anti-rabbit or anti-mouse secondary antibodies. Membranes were developed using Western Lightning Plus reagent (Perkin Elmer, Cat# NEL103001EA) and imaged on a Gel DocTM XR + Gel Documentation System.

## ISRE luciferase reporter gene assays

HEK-293T cells were seeded into wells of a 24-well plate and transfected 24 h later with pISRE-luciferase (Stratagene) (from which firefly luciferase is expressed under the control of a type I IFN-activated promoter) (250 ng/well), pRL-TK (Promega) (which constitutively expresses *Renilla* luciferase) (25 ng/well) and a pEGFP-C1 construct (250 ng/well) encoding GFP alone, CVS-P Δ30 (Δ30; P protein from which the C-terminal 30 amino acids are deleted, preventing binding to STAT1[27]), CVS-P3 WT, CVS-P3-KRm or CVS-P3-D289N. Transfection of plasmids was performed using Lipofectamine 2000 (Thermo-Fisher), according to the manufacturer's instructions. Cells were treated 7 h p.t. with or without 1000 U/mL IFNα (PBL Interferon Source) for 16 h before lysis using passive lysis buffer (Promega). Luciferase activity in lysates in 96-well plates was measured by a dual luciferase assay[27,47] using a BMG CLARIOstar plate-reader with solution A for firefly luminescence (200 mM Tris–HCl, 15 mM MgSO4, 0.1 mM EDTA, 25 mM DTT, 1 mM ATP, 200 mM luciferin, pH 8.0) and solution B for *Renilla* luminescence (25 mM tetrasodium pyrophosphate (Na4PPi)), 10 mM sodium acetate (NaAc), 15 mM EDTA, 500 mM Na2SO4, 500 mM NaCl, 50 mM 4-(6-Methyl-1,3-benzothiazol-2-yl)aniline, 4 mM benzyl-coelenterazine, pH 5.0). Relative Luciferase Activity (a.u.) was determined by normalizing values for firefly luciferase activity to those for *Renilla* luciferase activity for each well[47]. The mean Relative Luciferase Activity was calculated from biological triplicates for each sample.

## Recombinant protein purification

CVS P1, wt P3, P3-KRm and P3-D289N proteins were recombinantly expressed and purified in BL21 (DE3) *E. coli* expression system. 6xHis-GB1 solubility tag with a TEV protease cleavage site was introduced into the N-terminal of the P protein DNA sequences (Uniprot ID:

P322363) and inserted into the pET15b vector (constructs synthesized by GenScript). The P protein CTD constructs (residues 185-297 of wt P3, KRm and D289N) only contained N-terminal 6x-His tag with a TEV cleavage site. The constructs were transformed into chemically competent *E. coli* BL21(DE3) cells and positive transformants were grown overnight in 2YT media at 37 °C in shaker incubator. 10 mL of the overnight grown culture was inoculated into 1 L 2YT media and allowed to reach an $OD_{600}$ of 0.6 at 37 °C in a shaker incubator. The culture was then cooled to 16 °C and protein expression induced with 1 mM IPTG for 16 h. Cells were then harvested by centrifugation and stored at -20 °C. For constructs of P protein CTDs, $^{13}C,^{15}N$ labeling of wt CTD, the N5052 minimal medium was supplemented with $^{15}NH_4Cl$ and D-$[^{13}C]$ glucose (Sigma-Aldrich) as the sole source of nitrogen and carbon for the bacterial cells; expression was induced with 0.4 mM IPTG at $OD_{600}$ of 0.6-0.7 at 16 °C for 16 h.

For purification, cells were solubilized in 50 mL of buffer containing 50 mM $Na_2HPO_4$ at pH 7.4, 300 mM NaCl, 10 mM imidazole, 1 mM β-mercaptoethanol, supplemented with 10 μg/mL of RNase A (final concentration) and a complete PIC tablet (Roche). Cell pellets were lysed using an Avestin EmulsiFlex C3 cell crusher followed by centrifugation at 4 °C, 13,000 × *g* for 40 min to remove insoluble cell debris. Proteins were purified from the soluble fraction by affinity chromatography over Talon Superflow resin (Takara Clontech), which was equilibrated with 50 mM $Na_2HPO_4$, 300 mM NaCl, 1 mM β-mercaptoethanol, and 10 mM imidazole at pH 7.4, and were eluted with 350 mM imidazole in 50 mM $Na_2HPO_4$, 300 mM NaCl, 1 mM β-mercaptoethanol at pH 7.4. Eluted protein was exchanged into equilibration buffer and the His$_6$-GB1 tag cleaved overnight by incubating with TEV protease at 4 °C. Cleaved protein was then purified by reverse-affinity chromatography using Nickel-NTA resin (Takara) equilibrated in the same buffer. The flow-through and wash fractions containing cleaved protein were concentrated to 9 mL using 10-kD MWCO Amicon concentrators, before loading onto a Superdex S75 column (GE Healthcare) equilibrated in SEC buffer (50 mM $Na_2HPO_4$, 100 mM NaCl, 1 mM TCEP at pH 7.4) for size exclusion chromatography.

## RNA binding of recombinant proteins
Proteins were purified as above, but in the absence of RNAse A. 100 μg/mL protein was loaded onto a 1.2% Agarose gel and electrophoresis performed in 1x TAE buffer (40 mM Tris, 1 mM EDTA, 20 mM glacial acetic acid) with 0.5x SYBR-safe (Invitrogen™) at 140 V for 25 min, followed by imaging on a ChemiDoc™ imaging system equipped with Image Lab™ software. For RNase treatment, protein samples were incubated with 3 μL of RNAse A (10 mg/ml) supplied by Cell Signaling Technology SimpleChIP®(Cat#7013) for an hour at 4 °C before loading.

## Small-angle X-ray scattering
Small-angle X-ray scattering (SAXS) measurements were conducted at the Australian Synchrotron SAXS/WAXS Beamline equipped with a co-flow system to avoid radiation damage and enable higher X-ray flux (11,500 eV) and with an in-line size exclusion chromatography (SEC) to limit protein sample dilution (1–3). 50 μL of purified CVS P1, wt P3, P3-KRm and P3 D289N (5 mg/mL), were injected over a precision Superose 6 5/150 increase column (GE Healthcare) equilibrated and eluted with 25 mM $Na_2HPO_4$, 150 mM NaCl buffer (pH 7.4) containing 1 mM TCEP (tris(2-carboxyehtyl)phosphine) and 0.2% sodium azide. The sample-to-detector length used was 3256 mm, providing a q range of 0.007 to 0.515 Å$^{-1}$. Collected SAXS data was reduced using the Scatterbrain software, analyzed by CHROMIXS[93], and the ATSAS 3.2.1 software package[94]. SAXS patterns, the radius of gyration ($R_g$), the maximal particle dimension ($D_{max}$), and the pairwise distance distribution histogram [P(r) plot] were analyzed by using the ATSAS software suite. We used Ensemble Optimization Method (EOM)[53,54] to investigate the flexibility of the disordered regions in the P proteins. A homology

model of the dimerization domain of CVS P protein was prepared from the crystal structure of the same domain from the China/MRV strain (3L32), and the crystal structure of the C-terminal domain (1VYI) was used as input for the fixed core region of the proteins. RANCH program within EOM 2.0 was used to generate a pool of 10,000 models based on the P protein sequence, imposing P2 symmetry on the core dimerization and C-terminal domains, but not IDR1 and IDR2, and the average theoretical intensity was calculated using Crysol. *GAJOE* the genetic algorithm tool was used to compare the averaged theoretical scattering intensity from the pool of 10,000 conformations against the recorded scattering data for the proteins. The models that best-fit to the recorded scattering data were selected from the pool of each P protein as shown in Fig. 5. A summary of the SAXS data acquisition parameters is provided in Supplementary Table 1. The first 100 data points (before the void volume) were averaged as buffer scattering data and subtracted from the corresponding protein scattering data.

## Quantitative crosslinking-mass spectrometry
Purified CVS P1, wt P3, P3-KRm and P3-D289N samples (0.1 mg/mL) in 50 mM $Na_2HPO_4$, 150 mM NaCl at pH 7.4 were mixed with 100-fold access of disuccinimidyl sulfoxide (DSSO) crosslinker[56] (cross-links lysine residues within 10.1 to 30 Å) dissolved in DMSO, followed by gentle shaking for 30 min at RT. The crosslink reaction was terminated by adding Tris buffer at pH 8.0 to a final concentration of 20 mM in each sample. Trypsin enzyme was added to the samples to digest the crosslinked proteins and were left in the incubator shaker overnight at 37 °C overnight. Digested samples were then subjected to solid phase extraction using an SPE cartridge (Oasis HLB). Crosslinked peptides were eluted in 80% acetonitrile with 0.1% TFA followed by freeze drying. The freeze-dried samples were reconstituted in 2% acetonitrile containing 0.05% trifluoroacetic acid before injection onto the mass spectrometer. Four biological replicates were performed for each protein tested.

LC MS/MS was carried out using an Orbitrap Ascend mass spectrometer (Thermo Fisher, USA). The LC system was equipped with an Acclaim Pepmap nano-trap column (Dinoex-C18, 100 Å, 75 μm × 2 cm) and an Acclaim Pepmap RSLC analytical column (Dinoex-C18, 100 Å, 75 μm × 50 cm). The tryptic peptides were injected into the enrichment column at an isocratic flow of 5 μL/min of 2% v/v acetonitrile containing 0.1% v/v formic acid for 6 min, applied before the enrichment column was switched in-line with the analytical column. The eluents used for the LC were 5% DMSO/0.1% v/v formic acid (solvent A) and 100% CH$_3$CN/5% DMSO/0.1% formic acid v/v. The flow gradient was (i) 0-6 min at 3% B; (ii) 6-35 min, 3-22% B; (iii) 35-40 min, 22-40% B; (iv) 45-50 min, 40-80% B; (v) 50-55 min, 80-80% B; (vi) 55-56 min 85-3% and equilibrated at 3% B for 10 min before injecting the next sample.

The mass spectrometer was operated in positive-ionization mode with spray voltage set at 1.9 kV and source temperature at 275 °C. Survey scans (MS1) were recorded at 60,000 resolution on scan range from 375 to 1600 m/z with standard AGC and automated injection time. Charged states of 3-8 were selected for fragmentation as cross-linked peptides tend to occur at a charge state of 3+ or above. Dynamic exclusion of 30 second was applied and the precursor selection was prioritized based on higher charge state. To identify the cross-linked peptides, the MS2CID-MS3CID (MS2-MS3) workflow is used. The first MS/MS fragmentation was carried out using CID with collision energy of 25% and 10-ms CID activation time. The detection was in the orbitrap analyzer at 30,000 resolution with maximum injection time of 70 ms. The two most abundant reporter doublets from the first MS/MS scans (charge states 2 − 6, Δ-mass 31.9721 Da and mass tolerance of ±30 ppm) were selected for MS3 fragmentation. MS3 scans were recorded in the ion trap operated in rapid mode. Fragmentation was carried out using CID mode with 35% NCE, 10-ms activation time, activation Q of 0.25,

maximum injection time of 120 ms and normalized AGC target of 200%.

Cross-linked peptides were identified using the XlinkX node[57] implemented in Proteome Discoverer 2.3 (Thermo Fisher Scientific) together with the Sequest search engine as described previously[26]. Quantitation was carried out using the Minora feature detector and Precursor ion quantifier nodes (Top N algorithm used, $N = 3$) in Proteome Discoverer. As these samples comprised single purified proteins of the same concentration no normalization or scaling was carried out. Search parameters are as follows, MS1 accuracy: 10 ppm; MS2 accuracy: 20 ppm; MS3 accuracy: 0.5 Da. Database search was carried out using a user defined database with the sequences for phosphoproteins, P1, P3, P3-D289N and P3-KRm, from the CVS strain. Static modification detected is the carbamidomethylation (cysteine, +57.021 Da) and the dynamic modifications are hydrolyzed DSSO (lysine, +176.014), Tris DSSO (lysine, +279.078), oxidation (methionine, +15.995) and acetylation of the protein N-terminus (+42.011). The false discovery was set to be 1% from the XlinkX/PD validator. To reduce the number of false discoveries for the cross-linked peptides, XlinkX scores were filtered for an identification score of $\geq 50$. Peptides that showed abundance in at least three out of four replicates were considered for further analysis. Peptide abundances obtained for each replicate of P1, P3-D289N and P3-KRm were compared against P3 abundance values and the abundance ratios for P1/P3, P3-D289N/P3 and P3-KRm/P3 were calculated using the pairwise-ratios approach in Proteome Discoverer software. $P$-values for each peptide abundance ratio were calculated in GraphPad Prism using a two-tailed t-test (Wilcoxon test) by comparing the median abundance ratio values to 1. Any peptide identified in only P1, P3-D289N or P3-KRm but not in P3 and any peptide with abundance ratio greater than 40 was considered 100% abundant in the particular P protein and assigned a ratio value 100 for quantification. Cross-link data were visualised using xiNET cross-link viewer (http://crosslinkviewer.org).

## NMR Spectroscopy

All protein samples were buffer exchanged into 50 mM $Na_2HPO_4$, 100 mM NaCl, 1 mM TCEP pH 6.8 and all experiments were collected at 293 K. NMR assignment of CVS P protein CTD was accomplished on $^{13}C$, $^{15}N$-labeled protein on data acquired on a Bruker 700 MHz Avance IIIHD NMR spectrometer, equipped with TCI cryoprobe, using Topspin 3.2.7. 2D $^{15}N$, $^{1}H$ HSQC (9091 Hz, 1024 complex points ($^{1}H$), 1986 Hz, 128 complex points ($^{15}N$), 16 scans per row) and triple resonance experiments HNCACB and HN(CO)CACB (9091 Hz ($^{1}H$), 1986 Hz ($^{15}N$), 12315 Hz ($^{13}C$)) were recorded for the protein backbone resonance assignments. All triple resonance NMR experiments were recorded with 1024 ($^{1}H$), 32 ($^{15}N$), 64 ($^{13}C$) complex points and 64 scans per row, using using 10% non-uniform Poisson Gap sampling and reconstructed using SMILE. All NMR spectra were processed using NMRPipe (vs10.9)[95] and analyzed in NMRFAM-SPARKY (vs1.47)[96]. The $^{1}HN$, $^{15}N$, $^{13}C\alpha$ and $^{13}C\beta$ assignment data for the wt CTD have been deposited at the BMRB, (https:\\bmrb.io) accession code 53374. A full $^{1}H$,$^{15}N$ spectrum of wt CTD is in Supplementary Fig. 15. Reassignment of resonances of CTD-KRm and CTD-D289N were made by comparison of 2D $^{15}N$, $^{1}H$ HSQC spectra.

Hydrogen-deuterium exchange of wt CVS CTD, CTD-D289N and CTD-KRm was monitored via acquisition of 2D $^{15}N$, $^{1}H$ HSQC spectra. Exchange was initiated by passing a 250 μM sample over an Illustra NAP-5 column pre-equilibrated in NMR buffer, 100% $^{2}H_2O$. NUS was used for acquisition with 25% sampling. For each spectrum 16 scans were acquired per $^{15}N$ data point resulting in acquisition times of 20 min. Acquisition of the first spectrum occurred after 13 mins following initial exchange. Exchange rates ($k_{ex}$) (Supplementary Table 2) were determined by fitting to a single exponential.

## Turbidity assay

20 μL purified protein in SEC buffer was incubated in equal volume of aggregation buffer (100 mM Tris pH 8, 300 mM NaCl, 20% PEG 6000) at 25 °C for 10 min in an Eppendorf tube. Turbidity of the sample was estimated by measuring the light depletion at 600 nm using a benchtop Eppendorf spectrophotometer with 1 mm optical path length. This assay was performed for each protein in triplicate with error bars representing S.D. of mean. For measurement of turbidity of proteins in the presence of torula yeast tRNA, 20 mM stock of the RNA was prepared in the SEC buffer. The concentration of RNA was measured spectrophotometrically and calculated using 9250 $M^{-1}$ $cm^{-1}$ as the molar extinction coefficient, expressed as molarity of phosphate groups[97].

## Centrifugation-based Csat measurement

20 μL 200 μM protein was incubated with an equal volume of aggregation buffer as described in the turbidity assay. Following incubation, samples were centrifuged for 30 min at $16,100 \times g$ in a benchtop centrifuge. A gel-like pellet (dense phase) was observed at the bottom of the tube after centrifugation with a clear supernatant above (light phase). The supernatant was transferred to a fresh Eppendorf tube and protein concentration estimated by measuring absorbance at 280 nm. For Csat quantification of protein in the presence of RNA, the required concentration of RNA was optimized by measuring the $A_{260}/A_{280}$ ratio of the light phase at different molar ratios of protein:RNA (1:1–1:0.05). The RNA-protein ratio that produced a light phase $A_{260}/A_{280}$ ratio between 0.6–0.7 was used for experiments to minimize the contribution of RNA to protein quantification using $A_{280}$ measurements.

To monitor RNA binding and sequestration into phase-separated protein droplets, an equimolar concentration of RNA to protein was added to the P protein samples before incubating in aggregation buffer for 10 min at 25 °C. The samples were then centrifuged at $16,100 \times g$ for 30 min to separate the light and dense phases. The light phase was decanted and pelleted dense phase resuspended in 10 μL SEC buffer. Samples from the two separated phases were then analysed on a 1% agarose gel stained with SYBR Gold to detect RNA.

## Fluorescent labeling of proteins

CVS P1, P3, P3 D289N and P3 KRm were mutated to substitute the C-terminal residue Ser297 to Cys (S297C) to enable labeling with BODIPY R6G maleimide, a thiol-reactive fluorescent dye (Ex/Em = 530/548 nm). A 5-fold molar excess of dye was added to the purified proteins in SEC buffer (PBS pH 7.4, 2 mM TCEP) and incubated overnight at 4 °C in the dark. The reaction was quenched by adding an excess of β-mercaptoethanol, and labeled proteins were purified by SEC on a Superose 6 10/300 column equilibrated in SEC buffer.

## CLSM of in-vitro protein LLPS

BODIPY-labeled purified proteins were mixed with aggregation buffer at protein concentrations 12.5 μM, 25 μM, 50 μM and 75 μM for 10 min at 25 °C. 14 μL of each sample was transferred to a glass microscope slide (Trajan) and covered with a glass coverslip (Menzel Gläser). Samples were imaged at room temperature using a Zeiss Elyra LSM 880 inverted microscope with Plan-Apochromat 40x objective and 1 Airy unit pinhole. Z-stack images were acquired at 0.5–1 μm intervals using a 488 nm laser line. The area imaged was 212.5 × 212.5 μm in the x-y axis. Three images representing different regions of each glass slide were acquired for all four concentrations tested. All samples were imaged within 20 min of protein mixing with the aggregation buffer.

Images were processed and analyzed using Fiji (ImageJ) software. All images in the stack were first combined into a composite image using the z-project function. To determine the number of droplets and particle radii, the in-built particle analyzer algorithm was used. The detected particle size limit was set to 0.1- 400 μm²

and the area of droplets calculated. The areas were then converted to apparent radii assuming that all fluorescent droplets observed in the image were spherical. This method only gives an apparent radius and may differ from real radius as several actively merging droplets were oblong shaped and have been assumed to be spherical for calculation.

## Structure prediction of P proteins and model validation

Structures of CVS P1, P3 homodimers and CVS P3 homodimer complexed with torula yeast tRNA (CVS P3-tRNA) were predicted using AlphaFold3[98] (alphafoldserver.com) under default settings. The protein sequences were derived from Uniprot[99], identifier P22363. The torula yeast tRNA[Ile] sequence was chosen for prediction of CVS P3-tRNA complex, tRNA[Ile] sequence was taken from literature[100]. Five structural models were generated in all three cases and the models were compared on the basis of their pTM scores. The best models for CVS P1, CVS P3 and CVS P3-tRNA were searched on the Foldseek structural alignment tool server[101] to determine reference structures for the protein domains. The reference structures were further aligned to the best predicted models using US-align server[102] to obtain TM-scores.

## Statistical Analysis

Statistical analyses were performed using GraphPad Prism (v10) unless otherwise specified. The number of independent biological replicates (n) is indicated in each figure legend. No formal statistical methods were used to predetermine sample size; replicate numbers were chosen based on standard practice in the field to ensure reproducibility and sufficient power to detect meaningful differences. Data are presented as mean ± SEM unless otherwise stated. Normality was assessed where appropriate, and unpaired two-tailed $t$-tests (with Welch's correction for unequal variance) or one-way ANOVA with appropriate multiple-comparison correction were applied as indicated in figure legends. Exact $p$ values are reported in the figures or legends wherever possible, and significance thresholds are defined in each figure. All key findings were successfully replicated in ≥2 independent experiments, and representative images or data are shown where noted.

## Reporting summary

Further information on research design is available in the Nature Portfolio Reporting Summary linked to this article.

## Data availability

The XL-MS data and IP-MS have been deposited to the ProteomeXchange Consortium via the PRIDE[103] under accession codes PXD064535 and PXD064752. The NMR assignments of $^{15}$N, $^{1}$HN, $^{13}$Cα, and $^{13}$Cβ of CVS wt CTD have been deposited at the Biological Magnetic Resonance Data Bank (BMRB) under accession code 53374. Small-angle X-ray scattering (SAXS) data have been deposited in the Small Angle Scattering Biological Data Bank (SASBDB) under accession codes SASDRB6 (P1), SASDRX4 (P3), SASDRZ4 (P3-KRm), and SASDRY4 (P3-D289N). The AlphaFold models generated in this study have been deposited in ModelArchive under accession codes ma-1uvu0 (https://www.modelarchive.org/doi/10.5452/ma-1uvu0), ma-z7k76 (https://www.modelarchive.org/doi/10.5452/ma-z7k76), and ma-bpb4w (https://www.modelarchive.org/doi/10.5452/ma-bpb4w). The following Protein Data Bank (PDB) structures were used in this study: 8FUQ (CVS dimerization domain), 1VYI (CVS C-terminal domain), and 3L32 (China/MRV strain dimerization domain). Source Data are provided as a Source Data file. Source data are provided with this paper.

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

## Acknowledgements

This work was supported by National Health and Medical Research Council (NHMRC) project Grants 1125704 to G.W.M., P.R.G., 1160838 and 1079211 to G.W.M.; Australian Research Council grants DP210100998 to P.R.G., G.W.M., DP170104477 to T.D.M.B, DP150102569 to G.W.M.; and a Grimwade Fellowship provided by the Meigunyah Fund to G.W.M. We acknowledge the use of the biological SAXS beamline of the Australian Synchrotron, part of the Australian Nuclear Science and Technology Organization and the use of the NMR and Mass Spectrometry facilities at the University of Melbourne. We thank A/Prof. Nichollas Scott (The Peter Doherty Institute, Melbourne) for technical advice and assistance with mass spectrometry processing and analysis. We also acknowledge the Biological Optical Microscopy Platform (BOMP), University of Melbourne, and Monash Micro Imaging (MMI), Monash University, for their support and assistance with protein droplet imaging and confocal microscopy, respectively.

## Author contributions

SMR, SC, AS, PRG, and GWM conceived the study and designed the experiments. SMR, SC, AS, CTD, AMR, SU, KA, TZ, C-SA, ZHL, and FY performed the experiments. SMR, SC, AS, ARH, LEB, TZ, C-SA, ZHL, and FY analyzed the data. SC, AS, and PRG performed SAXS modeling, and PRG and SC carried out AlphaFold structural predictions. SMR, SC, and PRG prepared figures and visualizations. SMR, SC, PRG, and GWM wrote the original manuscript draft, and all authors reviewed and edited the

manuscript. SMR, GWM, PRG, TDMB, NAW, SA, and VS provided supervision and project oversight. SMR, GWM, and PRG administered the project. SO, DAJ, GWM, and PRG provided key resources and funding. GWM and PRG jointly supervised this work and contributed equally as senior authors.

## Competing interests

The authors declare no competing interests.

## Additional information

[1]Department of Microbiology, Biomedicine Discovery Institute, Monash University, Clayton, VIC, Australia. [2]Department of Biochemistry and Pharmacology, University of Melbourne, Parkville, VIC, Australia. [3]Bio21 Molecular Science and Biotechnology Institute, University of Melbourne, Parkville, VIC, Australia. [4]Australian Nuclear Science and Technology Organisation, Australian Synchrotron, 800 Blackburn Rd, Clayton, VIC, Australia. [5]Department of Biochemistry and Molecular Biology, Biomedicine Discovery Institute, Monash University, Melbourne, VIC, Australia. [6]Department of Microbiology and Immunology at the Peter Doherty Institute for Infection and Immunity, The University of Melbourne, Melbourne, VIC, Australia. [7]School of Chemistry, Monash University, Clayton, VIC, Australia. [8]Department of Anatomy and Developmental Biology, Biomedicine Discovery Institute, Monash University, Melbourne, VIC, Australia. [9]European Molecular Biological Laboratory Australia (EMBL Australia), Monash University, Clayton/Melbourne, VIC, Australia. [10]Commonwealth Scientific and Industrial Research Organisation (CSIRO), Australian Centre for Disease Preparedness (ACDP), East Geelong, VIC, Australia. [11]School of Medicine, Deakin University, Geelong, VIC, Australia. [12]These authors contributed equally: Stephen M. Rawlinson, Shatabdi Chakraborty, Ashish Sethi. ✉e-mail: stephen.rawlinson@monash.edu; prg@unimelb.edu.au; greg.moseley@monash.edu

