## [Transparent Peer Review file · Nature Communications]

Conformational dynamics, RNA binding, and phase separation regulate the multifunctionality of rabies virus P protein

Corresponding Author: Dr Gregory Moseley

Version 0:

Reviewer comments:

Reviewer #1

(Remarks to the Author)

The manuscript "Definition of mechanisms of viral protein multifunctionality reveals key roles of 'conformer diversity' and RNA interaction in forming host interfaces" by Rawlinson et al. aims at rationalizing the different biophysical and cellular behaviour of proteins P1 and P3 (natural truncation of P1) from Rabies virus. The authors perform a very complete study on the cellular localization of these two proteins and several mutants, concluding that both proteins present a different capacity to associate with several known MLO. Interestingly, P3 presents a gain of function, despite that it is a truncated form of P1. In vitro, both proteins can form condensed phases (in the presence of PEG) but differ in the capacity to interact with RNA. The authors rationalize these observations based on a complex network of intramolecular interactions that are perturbed upon the truncation of the N-terminal part of the protein. This hypothesis is (partially, in my opinion) supported by SAXS and XL-MS experiments.

The cellular experiments analysing the localization of the different protein forms seems solid to me, although I have to admit that this is not my area of expertise.

I have some major concerns:

- The concept that the different phenotype of viral products of the same gene arises from a distinct conformational behaviour is very interesting. However, this hypothesis has been already reported in a previous article of the same group (Sethi, A. et al. PNAS 2023) where two forms of P3 protein from a different strain of the rabies virus were studied using similar approaches. In this sense, I have doubts about the conceptual novelty of this manuscript.

- Related. It is unclear to me that the authors demonstrate that P1 and P3wt have different conformational behaviour. Obviously, SAXS-derived parameters R_g and D_{max} of P1 are larger than these of P3. Therefore, it is difficult to justify that the N-terminal part of P1 is compact. Moreover, the number of connections (cross-links) from the N-terminal part of P1 to the rest of the protein is not very large. In this context, it is difficult to provide a structural perspective to the different behaviour (specially the different affinity for RNA) of both proteins as the authors have done. Note that the authors have demonstrated the structural difference between the three P3 forms. The study of P1 constructs hosting the two mutations described from P3 (D289N and KRm) could be useful to provide a structural model of the difference between both isoforms.

- It is unclear how the dimeric arrangement of the protein affects the compaction (and changes in compaction) of the protein. The authors should describe how the dimer looks like and try to explain XL-MS data with the dimer arrangement. Is it possible to know whether the cross-links arise from the same chain or from two chains of the dimer? Would isotopic labelling be useful in this context?

- Structural analyses. There are many aspects that should be clarified.

(1) Given that all the constructs studied are flexible, the application of ab initio and rigid-body modelling is not appropriate and should not be neither shown nor discussed in the manuscript. These two (single-conformation) modelling approaches are not suitable for proteins experiencing important conformational fluctuations. Only raw-data analyses and EOM results should be used and discussed. The main conclusions of this section: (i) enhanced compactness of D289N and (ii) enhanced extendedness of KRm mutant can be explained with the raw data, EOM analyses and XL-MS. No need for ab initio and rigid body modelling.

(2) Details of the EOM analyses should be provided in the M&M section. Parameters such as the number of conformations used for the pool and the genetic algorithm, and whether symmetry was used to build the pool of conformations should be explained. Note that symmetry should not be used to build ensembles of disordered proteins. All this information should be provided in Supplementary table 2.

(3) It should be clarified the modelling approach used for the fits displayed in figure 5.

(4) The SEC-SAXS profiles of the four proteins should be provided to verify the quality of the sample.

Reviewer #2

(Remarks to the Author)

The manuscript by Rawlinson et al. investigates how different protein isoforms of the P gene of the rabies virus have different subcellular localization and engage different molecular interactions due to their different higher order organization. The concept of this paper is interesting but my major concern is that the the localization experiments and the interactome analysis were performed upon overexpression of the viral proteins. This is particularly relevant since many of the described interactions are triggered by phase separation, so that protein concentration should be considered.

Major points

As a more general suggestion I would include perturbation experiments such as treatment with LLPS inhibitors, RNaseA, Actinomycin D which leads to the segregation of the nucleoli in different components (ref.) to assess the effect of nucleolar stress and RNA on the localization of the different P protein isoforms.

Figure 1 B & 1E. The micrograph showing GFP-P3 localization looks quite unhealthy with disaggregated nucleoli. This panel is supposed to show that Taxol treatment affect interactions with microfilaments. The authors do not comment on the effect on the nucleolus. Is this just an healthy cell?

Figure 1F. Upon LMB treatment the nuclear foci formed by the WT protein increase in size. Could this be due to increase LLPS because of the increased nuclear concentration of the protein? What is the relevance of these nuclear foci in physiological conditions?

Figure 2B. The nuclear foci formed by GFP-P3 localize closed to several known nuclear bodies but do not colocalize with any of them. Again I wonder what is the functional meaning of this observation and if the behavior of the GFP-P3 proteins really reflects the protein produced during viral infection.

Reviewer #3

(Remarks to the Author)

Based on their studies with N-terminally truncated rabies virus (RABV) phosphoprotein P3, the authors claim to define 'mechanisms of viral protein multifunctionality' with 'conformer diversity' and RNA interaction in forming host interfaces. Basically, this means that multifunctionality of the phosphoprotein is not solely determined by the various truncated versions translated by leaky scanning, but also by conformational diversity of the various P forms by intra-protomer interactions. Simplified, by 'open' and 'more circular' conformations of a protomer, binding sites for interaction with other proteins and thus functionalities are either hidden or available. The authors claim this a novel mechanism to add multifunctionality to the limited virus proteome. The work is mainly characterized by mutational analysis of P3 (with gain and loss of function phenotypes for binding to membrane less organelle (MLOs) usually generated by liquid-liquid-phase-separation (LLPS). By SAXS and XL-MS analysis, wt and mutant P protein were structurally characterized and conformations were correlated with functions (MLO association / LLPS). Even though mutants lost MLO association, they not lost the ability to form LLPS structures in vitro by themselves. From these and other observations observations the authors conclude that specific inter-molecular interactions enable MLO targeting. The main argument for the major conclusion is, that both full length P1 and truncated P3 interact with MLO-associated proteins, whereas only P3 was able to bind RNA, which was positively or negatively affected by mutations that correlated with enhancement or inhibition of a closed conformation.

This very complex manuscript is excellently written and except some minor points/questions (see below) the presented data support drawn conclusions. The combination of different high end analytical techniques to correlate structural informations with functional data is exceptional.

However, although the manuscript describes high level scientific work about the mechanism how P isoforms mechanistically achieve multifunctionality and thus may support virus replication, I am not sure about the 'novelty' regarding the role of structural diversity as a driver of different protein functions. It is wide-spread phenomenon that proteins can achieve different conformations, sometimes leading to shifts in functionality. I recommend to alter the term 'conformer diversity' to 'conformational diversity'. Or is it an already defined term? I also recommend to dampen the title and discussion a bit and to more highlight the impact of this work on the understanding of the mechanisms how multiple conformations of P isoforms determine RABV biology than to claim a novel conformer diversity mechanism.

Specific points:

164: Taxol experiment for P1 requires photobleaching of soluble GFP in cytoplasm. The presented images are not suitable to microtubule stabilization/stabilization because of abundant levels of soluble GFP in the cytoplasm. In any case, confirm microtubules in both P1 and P3 expressing cells by immunostaining.

188/189: Rephrase. 'presence' is not an 'activity'

Figure 1: why do the authors exclusively use GFP-tagged P proteins? If conformation is the critical issue, it would be helpful to see the phenotypes in absence of any tag. Since only static ratios have been quantified, there is no need for live imaging

and IF could be done.

37/38: 'Despite reduced MLO association of P1 and P3-loss-of-function mutants compared with wild-type P3, they retain capacity for LLPS in vitro' vs. 223/224 'Our finding that P3 can undergo LLPS in vitro, and that this is impaired by mutations that impair MLO interactions indicate that an intrinsic capacity for LLPS is important': Do P3-loss of function mutants retain or lose LLPS capacity? This is not clear to me when I consider these two sentences. Check also conformity with later parts.

230 and (Figure 2A): To prove fusion longer observation and different z-planes have to be considered. It could simply demonstrate overlapping particles in different z-planes, although fusion is most alike.

Reviewer #4

(Remarks to the Author)

This paper describes a comprehensive study of a truncated isoform of the P gene of rabies virus, P3 and its mutants to understand the molecular basis underlying loss/gain-of function. The results have suggested mutation-induced interaction changes of various P forms with MLO proteins and within themselves. Importantly, it was found that P3 mutants have differential RNA binding, directly correlating with loss/gain-of function. While the content is interesting, the molecular/structural details underlying P3 RNA binding and functions remain elusive. Without quantitative analysis, it is difficult to determine whether P1 and P3 bind to the same and/or different MLO proteins, and whether P3 mutations induce conformational changes associated to its functions. The manuscript will be strengthened if specific interaction and conformational changes can be directly linked to P3 functions and a structural model can be derived to provide new mechanistic insights. Here are some specific points that need clarification:

1. It was claimed that P3 proteins associated with multiple MLOs as shown in Figure 4, it remains unclear whether the listed proteins are specific interactors to P3 as most of the proteins identified are abundant proteins in cells such as DDX proteins, Heat shock proteins, ribosomal proteins, HNRNP proteins, etc. In addition, the selected proteins appear to interact with both P1 and P3. Quantitative MS analysis would be needed to determine differential binders to each P proteins. Moreover, it would be more informative if cross-linking mass spectrometry could be applied to determine the direct interactors of P3 proteins. This will provide direct evidence for understanding interaction-dependent functions. Furthermore, co-localization of the selected interactors with P3 proteins could be performed to validate the identified interactions.
2. As shown in Figure 6, the overall intra-protein interactions revealed by cross-linking mass spectrometry appear very similar among P1 and different P3 forms. The absence of certain cross-links could be caused by structural changes; however, it could be due to variance in sample preparation and data-dependent acquisition during MS analysis. How many biological replicates were performed and what is the reproducibility? Quantitative XL analysis would be needed to determine whether and how deletion and mutations impact these intraprotein interactions within different P proteins.
3. It is unclear whether the cross-link data correlates well with SAXS data to support the conformational hypothesis. In addition, it is unclear what intra-protein and/or inter-protein interactions are responsible for loss of function in P3-KRm mutant, and gain of function in P3-D289N. It would be helpful to derive a structural model that can satisfy experimental data and describe a structural basis underlying P3's loss/gain of function and their differential interactions with RNAs.
4. In Supplemental table 1, the meaning of the assay needs to be described. It is unclear which protein was used as the bait. Details on protein identification and specific interactors should be included. How many biological replicates were performed for each bait? What is the FDR for protein and peptide identification?
5. Details on cross-link identification needs to be provided, including the number of cross-linked spectrum matches, unique cross-linked peptides, unique K-K linkages, data reproducibility and FDR.
6. All mass spec raw data (IP-MS, XL-MS, HDX) need to be deposited in public repository.

Version 1:

Reviewer comments:

Reviewer #1

(Remarks to the Author)

The revised version of the manuscript "Conformational dynamics, RNA binding, and phase separation regulate the multifunctionality of rabies virus P protein" has notably improved, specially from a quantitative perspective. My concerns, mainly related with the structural characterization, have been properly addressed, and the new version is clearer.

I still have some remarks that I think that will improve the structural part of the manuscript.

Line 470: Dmax is the maximum intramolecular distance and not the particle diameter

Line 483: When describing the conformations of P3 (Fig 5H), one has to imagine a continuous of states going from highly compact to highly extended structures. The use of two states as the authors do is not justified. In general, it is hard to say which protein is more compact when their lengths are different. In my opinion, P3 seems more compact than P1 if one compares the experimental histograms with these of the pools. The clearest signal derived from SAXS data is that P3-

D289N is significantly more compact than the other P3 constructs (wt and KRm).

My main criticism with the new version refers to the overinterpretation of the AF models. These models can be used to suggest the presence of interactions between domains, but not to explain some of the cross-links (but not others). AF is very bad in predicting IDRs and, in any case, it represents proteins as a single conformation... but SAXS unambiguously tells the authors that P1 and P3 must be represented by very conformationally heterogeneous ensembles. So, I would be very cautious in interpreting the overall architecture of these models when we know that they are highly dynamic.

Reviewer #2

(Remarks to the Author)

I read the revised manuscript and the rebuttal letter addressing the comments. I am satisfied by the responses and with the modifications that the authors made to the manuscript. I do not have additional issues to raise. In my opinion the manuscript can be accepted for publication in Nature Communications.

Reviewer #3

(Remarks to the Author)

The revised manuscript adequately addresses all issues I raised in my former review.

Reviewer #4

(Remarks to the Author)

While the authors have addressed most of my previous concerns, a few remaining issues regarding result interpretation and data representation require further clarification.

1. Details on cross-link identification including the number of cross-linked spectrum matches, unique cross-linked peptides, unique K-K linkages, and data reproducibility are still missing. This information should be included in the text.
2. It is unclear how cross-linked peptides were quantified. Since there is no isotope label involved, how was the label-free quantitation performed? The data-dependent acquisition method described in the text cannot be used for accurate label-free quantitation. How were the quantitative values normalized?
3. The representation of the qXL data in Figure 6 is somewhat confusing. Quantitation should be shown for cross-linked residue pairs rather than individual residues. Additionally, the color annotations should be clearly explained, and each panel has to have labeled axes.
4. Each supplementary table should have clear and concise table titles. Details on protein identification are still missing in Table S1. Protein identification details including the number of identified and quantified peptides, identification scores, sequence coverage need to be included.
5. In Table S3, cells containing '#DIV/0!' or '0' should be replaced with blank spaces or a dash ('-'), as no quantifiable cross-links were present in these cells. Although the authors stated that four biological replicates were performed, only a single abundance value is reported for each cross-link. The experimental results from all replicates should be included and clearly described in the table.
6. Although the authors stated that cross-links for P1 and P3 support the predicted models in Figure 10. However, the C α -C α distances between cross-linked residues were not shown. These values should be included to illustrate whether the cross-links correlate well with the models. This information should be described in the text.

Version 2:

Reviewer comments:

Reviewer #1

(Remarks to the Author)

The authors have properly addressed my remarks and, in my opinion, the manuscript can be published.

Reviewer #4

(Remarks to the Author)

The authors have adequately addressed the remaining concerns.

Reviewer Comments (blue) and Responses (black)

Reviewer #1 (Remarks to the Author):

The manuscript “Definition of mechanisms of viral protein multifunctionality reveals key roles of ‘conformer diversity’ and RNA interaction in forming host interfaces” by Rawlinson et al. aims at rationalizing the different biophysical and cellular behaviour of proteins P1 and P3 (natural truncation of P1) from Rabies virus. The authors perform a very complete study on the cellular localization of these two proteins and several mutants, concluding that both proteins present a different capacity to associate with several known MLO. Interestingly, P3 presents a gain of function, despite that it is a truncated form of P1. In vitro, both proteins can form condensed phases (in the presence of PEG) but differ in the capacity to interact with RNA. The authors rationalize these observations based on a complex network of intramolecular interactions that are perturbed upon the truncation of the N-terminal part of the protein. This hypothesis is (partially, in my opinion) supported by SAXS and XL-MS experiments.

The cellular experiments analysing the localization of the different protein forms seems solid to me, although I have to admit that this is not my area of expertise.

I have some major concerns:

- The concept that the different phenotype of viral products of the same gene arises from a distinct conformational behaviour is very interesting. However, this hypothesis has been already reported in a previous article of the same group (Sethi, A. et al. PNAS 2023) where two forms of P3 protein from a different strain of the rabies virus were studied using similar approaches. In this sense, I have doubts about the conceptual novelty of this manuscript.

We thank the reviewer for raising this point for clarification.

In the PNAS paper, we proposed that functional differences between P3 proteins from a pathogenic and an attenuated rabies virus strain were due to differences in conformational states — specifically, that a single substitution (N226H) shifted P3 from a closed to an open conformation, affecting immune evasion, LLPS, and localization. These data were consistent with the concept that conformational diversity (i.e. the ability of the same protein to adopt multiple distinct structural states), could underlie viral protein multifunctionality. However, the PNAS paper focused on strain-specific variation correlating altered functions of P3 with structural impacts; it did not define the mechanism by which P3 gains multiple functions compared with P1.

In this new study, we elucidate for the first time how P3 gains extensive functions compared with the longer P1 isoform, including identifying RNA binding (which was not identified or hypothesized in the previous work) as a principal defining mechanism. Specifically, we showed that:

- P1 and P3 differ not only through the loss of sequence in the latter, but also through extensive long distance conformational effects through distinct interactions of ordered and disordered regions.
- A critical element in the gain of function of P3 compared with P1 is RNA binding, which results from these conformational changes (**note:** this is new data following reviewer suggestions which has elucidated the molecular basis of RNA binding and its switching on-or-off between isoforms, a highly novel discovery)
- This conformer-dependent RNA binding governs interaction with host LLPS structures (e.g., nucleoli, nuclear bodies, microtubules), revealing how conformational diversity underpins the complex phenotype of P3.

Together, these data provide novel insights beyond that in the PNAS paper in particular showing multifunctionality of the viral protein derives from RNA binding, which is regulated due to long range coordinated conformational changes. This provides the first insight on how truncation of this viral protein results in extensive gain of function, highlighting novel mechanisms by which viruses expand functional capacity of limited genomes.

We thank the reviewer again for encouraging us to expand on this point and have modified the text (e.g. **lines 137-150 and Abstract**) to clarify on this.

- Related. It is unclear to me that the authors demonstrate that P1 and P3wt have different conformational behaviour.

Obviously, SAXS-derived parameters Rg and Dmax of P1 are larger than these of P3. Therefore, it is difficult to justify that the N-terminal part of P1 is compact. Moreover, the number of connections (cross-links) from the N-terminal part of P1 to the rest of the protein is not very large. In this context, it is difficult to provide a structural perspective to the different behaviour (specially the different affinity for RNA) of both proteins as the authors have done. Note that the authors have demonstrated the structural difference between the three P3 forms. The study of P1 constructs hosting the two mutations described from P3 (D289N and KRm) could be useful to provide a structural model of the difference between both isoforms.

We thank the reviewer for encouraging us to further investigate these important points. To address the differences of P1 vs P3 we performed quantitative XL-MS (qXL-MS) (as requested by reviewer 4) to gain a more accurate comparison (**New Fig 6**). There are cross-links between IDR1 and IDR2 and from these regions to the CTD that are present in P1 and absent in P3 that support that P1 is more compact relative to P3. To address a structural model (and also in response to reviewer 4) we prepared Alphafold3 models (which previously was not permitted for viral proteins) (**New Fig. S10**). Due to the content of disorder there is low confidence in the overall model, but the structure of the ordered domains are retained. These models propose that the CTD is differently oriented in P1 and P3 (consistent with overall/long range conformational diversity underpinning distinct functions) and that the CTD of P1 interacts with IDR1, in a manner that blocks access to the surface around K214 and R260, the two residues that, when mutated, lose RNA binding. In P3, both the orientation of the CTDs and the access to these residues, are likely to enable RNA binding. Critically, several XL-MS crosslinks present for P1 and absent in P3 support these models (Fig. S10). As expected, P1 and P3 are conformationally heterogeneous so many cross-links are absent, especially for the disordered regions which are modelled with low confidence. RNA binding was also modeled by Alphafold3 (**New Fig. S10C**).

We have summarized these models in **New Figure S10** and modified/included new text in several results (**lines 505-527; lines 704-716**) and methods sections.

- It is unclear how the dimeric arrangement of the protein affects the compaction (and changes in compaction) of the protein. The authors should describe how the dimer looks like and try to explain XL-MS data with the dimer arrangement. Is it possible to know whether the cross-links arise from the same chain or from two chains of the dimer? Would isotopic labelling be useful in this context?

We thank the reviewer and agree with the idea that interprotomer interactions are possible or likely and may be important for compaction, and that several crosslinks must be interprotomer and not intraprotomer. Isotopic labelling would be advantageous, but we have thus far not succeeded in generating a dimer of mixed labelled subunits (eg $^{14}\text{N}/^{15}\text{N}$).

To emphasize that interprotomer interactions are likely we have added text in support (**lines 535-545**).

- Structural analyses. There are many aspects that should be clarified.

(1) Given that all the constructs studied are flexible, the application of ab initio and rigid-body modelling is not appropriate and should not be neither shown nor discussed in the manuscript. These two (single-conformation) modelling approaches are not suitable for proteins experiencing important conformational fluctuations. Only raw-data analyses and EOM results should be used and discussed. The main conclusions of this section: (i) enhanced compactness of D289N and (ii) enhanced extendedness of KRm mutant can be explained with the raw data, EOM analyses and XL-MS. No need for ab initio and rigid body modelling.

Based on the reviewer's suggestion we have removed the ab initio and rigid-body modelling from the manuscript. We now focus solely on the EOM results for the analysis (Fig. 5). Additionally, the references to rigid-body modelling have been removed from both the Materials & Methods section and Supplementary Table 2. The manuscript now discusses only the raw data, EOM analysis, and qXL-MS results to address the flexibility of the constructs and the observed structural features.

(2) Details of the EOM analyses should be provided in the M&M section. Parameters such as the number of conformations used for the pool and the genetic algorithm, and whether symmetry was used to build the pool of conformations should be explained. Note that symmetry should not be used to build ensembles of disordered proteins. All this information should be provided in Supplementary table 2.

Complete details are now provided in the M&M (**lines 1055-1080**), including numbers of conformations generated, algorithms and software. We have also clarified that symmetry was applied only to the dimerisation domain (DD), but not to the flexible regions in the EOM modelling. This information is provided in Supplementary Table 2.

(3) It should be clarified the modelling approach used for the fits displayed in figure 5.

We thank the reviewer for raising this potentially unclear point and have clarified in the figure legend that only EOM modelling was used for the fits displayed in **Figure 5**.

(4) The SEC-SAXS profiles of the four proteins should be provided to verify the quality of the sample.

The SEC-SAXS profiles for the four proteins have now been included in **Figure S9**, as requested, to provide additional verification of the sample quality.

Reviewer #2 (Remarks to the Author):

The concept of this paper is interesting but my major concern is that the the localization experiments and the interactome analysis were performed upon overexpression of the viral proteins. This is particularly relevant since many of the described interactions are triggered by phase separation, so that protein concentration should be considered.

We thank the reviewer for raising these points for consideration. We acknowledge the potential issues of using transfection rather than infection, but note that this approach was necessary to assess specific functions/localization of P3 as direct and specific detection for microscopy, IP *etc.* of P3 in infected cells is not possible. This is because P3 is one of five isoforms expressed by virus that differ only due to N-terminal truncations. P3 is shorter than P1, and so antibodies recognizing P3 also recognize P1 (and other isoforms) making isoform-specific detection by immunostaining unfeasible.

An additional issue associated with analysis of proteins associated with LLPS structures/MLOs is that fixation approaches can disrupt such structures, or may not enable efficient antibody penetration of MLOs for immunostaining (e.g., Irgen-Giorgio et al., *eLife*, 2022), so accurate immunostaining often requires the optimization of specific, typically “gentle” fixation methods. Due to specific biosafety requirements, we must adhere to limited, defined fixation/inactivation conditions. We have inserted text describing these limitation in the results section (**lines 282-292**).

Despite these constraints, multiple prior studies—and new data presented here—support the physiological relevance of our findings using transfected cells:

- (1) Prior infection studies using RABV with or without mutations known to impair P3-MT interaction and bundling confirmed that in infected cells, MTs undergo increased bundling, which is impaired by the mutations, consistent with data for cells expressing wt or mutant P3 alone (*Sethi et al., PNAS, 2023*)
- (2) Previous IP analysis of P protein from infected cells identified interaction with nucleolin (*Oksayan et al., J Virol, 2015*), consistent with our data for P1 and P3 alone.
- (3) P3 expression alone and RABV infection produced comparable effects on PML-associated NBs (*Blondel et al., Oncogene, 2002*).
- (4) Several proteins identified as P3 interactors in our study (e.g., RPL9, Staufen-1, and previously confirmed P3 interactors: Treacle, PML) have established roles in infection, including effects on replication and/or immune modulation.

In addition to the above, we have now performed microscopy analysis of cells infected by RABV CVS-11 and immunostained for P protein, detecting punctate nuclear structures (consistent with nuclear bodies) in infected cells but not in mock controls (**New Figure S4**). As P3 is the major isoform that can localize to the nucleus, and P1 is excluded from the nucleus, this indicates that the NBs formed incorporate P3, supporting our data for expression of P3 alone.

We have revised the manuscript to include this new data and to address the limitations of the transfection system, as well as to expand the discussion of prior and current evidence supporting the physiological relevance of our findings (**lines 439–454**).

Major points

As a more general suggestion I would include perturbation experiments such as treatment with LLPS inhibitors, RNaseA, Actinomycin D which leads to the segregation of the nucleoli in different components (ref.) to assess the effect of nucleolar stress and RNA on the localization of the different P protein isoforms.

We thank the reviewer for this excellent suggestion and have performed these assays (**new Figures S12A, S12B; lines 624–629**). Actinomycin D caused redistribution of P3 into nucleolar caps adjacent to UBF1-enriched caps, consistent with classical nucleolar segregation under transcriptional stress. In contrast, RNase A inhibited P3 localization to the nucleolus, suggesting that RNA integrity is essential for its nucleolar association. These findings strongly support a role for RNA and nucleolar architecture in regulating P3 localization.

Figure 1 B & 1E. The micrograph showing GFP-P3 localization looks quite unhealthy with disaggregated nucleoli. This panel is supposed to show that Taxol treatment affect interactions with microfilaments. The authors do not comment on the effect on the nucleolus. Is this just an healthy cell?

We thank the reviewer for raising this important point. We did not observe consistent effects on nucleoli across the population of cells with this phenotype being rare; we have now replaced the panel with a more representative image.

Figure 1F. Upon LMB treatment the nuclear foci formed by the WT protein increase in size. Could this be due to increase LLPS because of the increased nuclear concentration of the protein? What is the relevance of these nuclear foci in physiological conditions?

Figure 2B. The nuclear foci formed by GFP-P3 localize closed to several known nuclear bodies but do not colocalize with any of them. Again I wonder what is the functional meaning of this observation and if the behavior of the GFP-P3 proteins really reflects the protein produced during viral infection.

We thank the reviewer for raising these points for clarification. To assess effects of nuclear concentration on NB association, we quantified and compared nuclear P3 fluorescence with the extent of NB formation. The results (**new Figure S3**) show a clear correlation between nuclear concentration and NB formation, consistent with LLPS behavior (**lines 275–279**). To assess the formation of P protein enriched nuclear foci during infection, we immunostained RABV-infected cells detecting nuclear P-protein foci consistent with NBs (above and **new Figure S4**).

The functional role of P3 nuclear foci remains incompletely resolved, but several lines of evidence support biological significance, including findings that PML NBs are targeted by RABV P protein isoforms and modified during infection in a fashion consistent with that observed following P3 expression (*Blondel et al., Oncogene, 2002*). Given PML NBs' key role in antiviral defense, this possibly reflects a viral strategy for immune evasion. RABV infection also induces the lncRNA *NEAT1*, required for paraspeckle formation (*Saha et al., J Gen Virol, 2006*), consistent with mechanisms to modulate antiviral nuclear MLOs (these points are clarified in **lines 315–322**).

We observe heterogeneous colocalization where in some cases, P3 foci are adjacent to known NBs (as noted by the reviewer) but in others show partial or complete colocalization, or apparent fusion. This is likely due to the dynamic and diverse nature of NBs, and consistent with the dynamic liquid nature of P3 NBs (**new Fig. 2A**), and with a dynamic regulatory interaction with cellular NBs. Importantly, new IF analysis with NOLC1 (**new Figure S8**) also reveals that some P3 nuclear foci colocalize with NOLC1, highlighting functional and structural heterogeneity. We have revised the manuscript to include **new data (Fig. S8)** and text clarifying these points (**lines 302–313**), and thank the reviewer for encouraging us to make these changes.

Reviewer #3 (Remarks to the Author):

Based on their studies with N-terminally truncated rabies virus (RABV) phosphoprotein P3, the authors claim to define ‘mechanisms of viral protein multifunctionality’ with ‘conformer diversity’ and RNA interaction in forming host interfaces. Basically, this means that multifunctionality of the phosphoprotein is not solely determined by the various truncated versions translated by leaky scanning, but also by conformational diversity of the various P forms by intra-protomer interactions. Simplified, by ‘open’ and ‘more circular’ conformations of a protomer, binding sites for interaction with other proteins and thus functionalities are either hidden or available. The authors claim this a novel mechanism to add multifunctionality to the limited virus proteome. The work is mainly characterized by mutational analysis of P3 (with gain and loss of function phenotypes for binding to membrane less organelle (MLOs) usually generated by liquid-liquid-phase-separation (LLPS). By SAXS and XL-MS analysis, wt and mutant P protein were structurally characterized and conformations were correlated with functions (MLO association / LLPS). Even though mutants lost MLO association, they not lost the ability to form LLPS structures in vitro by themselves. From these and other observations the authors conclude that specific inter-molecular interactions enable MLO targeting. The main argument for the major conclusion is, that both full length P1 and truncated P3 interact with MLO-associated proteins, whereas only P3 was able to bind RNA, which was positively or negatively affected by mutations that correlated with enhancement or inhibition of a closed conformation.

This very complex manuscript is excellently written and except some minor points/questions (see below) the presented data support drawn conclusions. The combination of different high end analytical techniques to correlate structural informations with functional data is exceptional.

However, although the manuscript describes high level scientific work about the mechanism how P isoforms mechanistically achieve multifunctionality and thus may support virus replication, I am not sure about the ‘novelty’ regarding the role of structural diversity as a driver of different protein functions. It is wide-spread phenomenon that proteins can achieve different conformations, sometimes leading to shifts in functionality. I recommend to alter the term ‘conformer diversity’ to ‘conformational diversity’. Or is it an already defined term?

We thank the reviewer for the very positive and useful suggestions. Although conformational changes are well established in protein functionality, we suggest that long range conformational diversity enabling high multifunctionality as described here is less well understood, including its role in highly multifunctional viral proteins. However, our aim here is not principally to suggest that conformational diversity is novel, but rather to that isoform-specific conformational regulation and resultant RNA binding, provides a novel mechanism by which the rabies virus P protein achieves multifunctionality beyond domain composition alone. We have modified the text (**lines 729-745**) to highlight these points.

We have also replaced the term “*conformer diversity*” with “*conformational diversity*” throughout revised the title to: “**Conformational dynamics, RNA binding, and phase separation regulate the multifunctionality of rabies virus P protein**”, and modified the text in the Discussion to better reflect the novelty of the conformationally regulated RNA binding as a novel regulatory mechanism generating multifunctionality, as well as emphasizing the relevance of our findings to rabies virus biology, rather than the novel aspects of conformational diversity in line with the reviewer’s comments. We thank the reviewer again for encouraging us to make these changes to improve the clarity of our manuscript.

Specific points:

164: Taxol experiment for P1 requires photobleaching of soluble GFP in cytoplasm. The presented images are not suitable to microtubule stabilization/stabilization because of abundant levels of soluble GFP in the cytoplasm. In any case, confirm microtubules in both P1 and P3 expressing cells by immunostaining.

We thank the reviewer for these excellent suggestions. We have performed photobleaching experiments to reduce soluble GFP-P1 and GFP-P3 fluorescence (**new data, Figure S2B**) and conducted immunostaining for microtubules (**new data, Figure S2A**). Photobleaching of GFP-P1 resulted in global depletion of soluble GFP fluorescence with no filamentous structures becoming apparent, consistent with the absence of microtubule association. In contrast, photobleaching of GFP-P3 revealed clear filamentous structures post-bleach, consistent with microtubule association. Immunostaining further confirmed that GFP-P3, but not P1, colocalizes with microtubules, reinforcing our previous findings that P3, but not P1, interacts with and stabilizes microtubules. These new data confirm the phenotypic difference between P1 and P3 for microtubule interaction. We have revised the text to clarify these points (**lines 190-208**), and thank the reviewer again for these suggestions to improve the manuscript.

188/189: Rephrase. 'presence' is not an 'activity'

We thank the reviewer and have modified the text (see section **lines 227-233**).

Figure 1: why do the authors exclusively use GFP-tagged P proteins? If conformation is the critical issue, it would be helpful to see the phenotypes in absence of any tag. Since only static ratios have been quantified, there is no need for live imaging and IF could be done.

We thank the reviewer for raising this important point. GFP-tagged P proteins were used to enable live-cell imaging, which avoids potential artefacts introduced by fixation. This is particularly relevant when examining LLPS-related structures, where fixation has been shown to disrupt LLPS (Irgen-Giorgio et al., *eLife*, 2022), and we have observed for GFP-P3 that detection of nucleolar localization by immunostaining is compromised under certain fixation conditions, despite clear nucleolar localization in live cells.

However, we concur that validation of phenotypes observed using non-tagged proteins is important and so have included new images of immunostained cells expressing non-tagged P1 and P3 (**new Figure S1; lines 170-172**), which confirm P1 is diffusely cytoplasmic (consistent with previous data for tagged and untagged forms (e.g., Oksayan et al., *JBC*, 2012) and P3 localizes to the nucleus, nucleolus, and nuclear bodies, as expected. While microtubule association of untagged P3 is less evident under the fixation conditions used (which were selected to preserve nuclear and nucleolar structures), MT association localization has been previously demonstrated for untagged P3 (Brice et al., *Sci Rep*, 2016).

37/38: 'Despite reduced MLO association of P1 and P3-loss-of-function mutants compared with wild-type P3, they retain capacity for LLPS in vitro' vs. 223/224 'Our finding that P3 can undergo LLPS in vitro, and that this is impaired by mutations that impair MLO interactions indicate that an intrinsic capacity for LLPS is important': Do P3-loss of function mutants retain or lose LLPS capacity? This is not clear to me when I consider these two sentences. Check also conformity with later parts.

We thank the reviewer for identifying this important point of potential confusion. The first sentence refers to data in this current study, while the second refers to our previous study, which used different mutations. In our previous study, substitutions in P3 between two RABV strains (Nishigahara and Ni-CE) resulted in loss of both in vitro LLPS and cellular MLO association in Ni-CE, suggesting that P3 LLPS is essential for MLO interaction. In the current study, we identify different P3 mutations that disrupt MLO association without abolishing LLPS capacity *in vitro*. This has allowed us to make the novel observation that while intrinsic LLPS capacity is important, LLPS alone is not sufficient for cellular MLO targeting such that other features (particularly RNA binding) are required. We have revised the text (**lines 266-269; 602-629**) to clarify on these important points, and to ensure consistency across the manuscript. We thank the reviewer again for highlighting this ambiguity.

230 and (Figure 2A): To prove fusion longer observation and different z-planes have to be considered. It could simply demonstrate overlapping particles in different z-planes, although fusion is most alike.

We thank the reviewer for this excellent suggestion, and have performed new live-cell time-lapse imaging with 3D z-stack acquisition. This clearly demonstrates dynamic fusion events of P3 nuclear bodies across multiple z-planes, confirming that the observed coalescence is due to fusion, rather than projection artefacts/overlap in different focal planes (**revised Figure 2A, new Supplementary Movie 1, modified text, lines 271-279**).

Reviewer #4 (Remarks to the Author):

This paper describes a comprehensive study of a truncated isoform of the P gene of rabies virus, P3 and its mutants to understand the molecular basis underlying loss/gain-of function. The results have suggested mutation-induced interaction changes of various P forms with MLO proteins and within themselves. Importantly, it was found that P3 mutants have differential RNA binding, directly correlating with loss/gain-of function. While the content is interesting, the molecular/structural details underlying P3 RNA binding and functions remain elusive. Without quantitative analysis, it is difficult to determine whether P1 and P3 bind to the same and/or different MLO proteins,

and whether P3 mutations induce conformational changes associated to its functions. The manuscript will be strengthened if specific interaction and conformational changes can be directly linked to P3 functions and a structural model can be derived to provide new mechanistic insights. Here are some specific points that need clarification:

We thank the reviewer for raising these important points and have performed additional experiments including quantitative IP/MS, XL-MS, and structural modelling to answer the queries. These experiments have substantially advanced and/or validated our findings (see below).

1. It was claimed that P3 proteins associated with multiple MLOs as shown in Figure 4, it remains unclear whether the listed proteins are specific interactors to P3 as most of the proteins identified are abundant proteins in cells such as DDX proteins, Heat shock proteins, ribosomal proteins, HNRNP proteins, etc. In addition, the selected proteins appear to interact with both P1 and P3. Quantitative MS analysis would be needed to determine differential binders to each P proteins. Moreover, it would be more informative if cross-linking mass spectrometry could be applied to determine the direct interactors of P3 proteins. This will provide direct evidence for understanding interaction-dependent functions. Furthermore, co-localization of the selected interactors with P3 proteins could be performed to validate the identified interactions.

We thank the reviewer for these valuable suggestions. While certain proteins identified in our initial IPs are abundant, several of these (e.g., Hsp90 and RPL9) have already been validated as rabies P-protein interactors (Dalidowska et al., *Int J Mol Sci*, 2022; Li et al., *Virology*, 2016), supporting their functional relevance. P3 has also been shown to inhibit ribosome biogenesis (Rawlinson et al., *Traffic*, 2023), supporting a significance of ribosomal interactions. Moreover, these protein families (e.g. hnRNPs, RNA helicases) are widely targeted by viruses and contribute to replication, host shutoff, and immune evasion (see reviews, e.g., Winnard et al., *Antiviral Res*, 2025; Maceratessi et al., *Immunology*, 2024).

However, we concur with the reviewer on the importance of validating the interactions against controls, and comparing the interactomes of P1 and P3, using quantitative methods. We thus conducted new quantitative IP-MS assays to compare the interactomes of GFP-P1 and GFP-P3 with GFP to validate the specificity of ‘hits’, and to compare directly the interactome of P1 with that of P3 (**revised Figure 4, new Figure S6, S7, and Supplementary Table 1**). We found that GFP-P3 interactome was significantly enriched compared with the control (GFP) for MLO-associated proteins (including abundant proteins alluded to above) from nucleoli, stress granules, P-bodies, paraspeckles, and nuclear speckles. Comparison of P3 and P1 confirmed largely overlapping interactomes (249 and 207 total high-confidence proteins for P3 and P1, respectively), although quantitative analysis indicated 20 proteins were preferentially associated with P3, and 5 with P1.

The fact that most proteins interactors were common to P1 and P3 suggests that these interactions do not define localization. To explore this further, we performed co-localization studies as suggested by the reviewer (**new Figure S8**), showing P3 colocalizes with NPM1 and NOLC1, but P1 does not. Thus, it appears that P1 has the capability to bind these proteins but several of these are only post-lysis for IPs, when cellular compartmentalization is disrupted supporting that additional factors (particularly RNA binding, are involved in P3-specific localization). These new data are incorporated into the revised manuscript (major modifications to results, **lines 371-454**), and we thank the reviewer again for prompting these experiments that have strengthened our conclusion.

2. As shown in Figure 6, the overall intra-protein interactions revealed by cross-linking mass spectrometry appear very similar among P1 and different P3 forms. The absence of certain cross-links could be caused by structural changes; however, it could be due to variance in sample preparation and data-dependent acquisition during MS analysis. How many biological replicates were performed and what is the reproducibility? Quantitative XL analysis would be needed to determine whether and how deletion and mutations impact these intraprotein interactions within different P proteins.

We thank the reviewer for these suggestions to improve the data, and have repeated the XL-MS experiments quantitatively (qXL-MS). As detailed in the M&M (**lines 1082-1140**) we applied qXL-MS on four biological replicates of each protein. We report peptide abundances obtained for each replicate of P1, P3-D289N and P3-KRm and compare against P3 abundance values (**new Fig. 6, Table S3**). Further in Table S3 we include the calculated abundance ratios for P1/P3, P3-D289N/P3 and P3-KRm/P3. Full details of the experiments, software and analysis are in the M&M. This approach generated clear results for cross-links present in P1 and absent in P3 (and *vice*

versa), and similarly in the two mutants, supporting differences in the ensembles of the four proteins. We have modified the corresponding result sections for P1 vs P3 (**lines 487-500**) and for P3-D289N and P3-KRm vs P3 (**lines 530-565**).

3. It is unclear whether the cross-link data correlates well with SAXS data to support the conformational hypothesis. In addition, it is unclear what intra-protein and/or inter-protein interactions are responsible for loss of function in P3-KRm mutant, and gain of function in P3-D289N. It would be helpful to derive a structural model that can satisfy experimental data and describe a structural basis underlying P3's loss/gain of function and their differential interactions with RNAs.

We thank the reviewer for this suggestion, and have conducted AlphaFold3 modeling on P1, P3 and P3 bound to RNA. As expected, these models show a poor confidence for the modelling of the IDRs. However, there are distinct differences for the orientation of the CTDs of P1 and P3. Further for P1 IDR1 interacts with the CTDs supporting a more compact structure. Importantly the orientation of the CTD of P1 with respect to IDR1 shows the latter blocks access to the surface around K214 and R260 of the CTD, which are the two residues that when mutated lose RNA binding. In P3, both the orientation of the CTDs and the access to these residues is changed resulting in exposure of this site in the CTD, which is likely to enable RNA binding. Critically, several XL-MS crosslinks present for P1 and absent in P3, and *vice versa*, support these models. As expected, P1 and P3 are conformationally heterogeneous so many cross-links are absent, especially for the disordered regions which, as stated, are modelled with low confidence. RNA binding was also modeled by AlphaFold3 which suggests that the CTD binds the RNA. This interaction is dependent on the orientation of the 2 CTDs in P3 (which differs and so is not functional in P1), supporting that conformational differences underpin the RNA binding that contributes to gain of function.

We have summarized these models in **new Figure S10** and **new text (lines 505-527; 704-716)** as well as adding a new methods section.

4. In Supplemental table 1, the meaning of the assay needs to be described. It is unclear which protein was used as the bait. Details on protein identification and specific interactors should be included. How many biological replicates were performed for each bait? What is the FDR for protein and peptide identification?

We thank the reviewer for these suggestions and have included all relevant information in the revised Materials and Methods section, including the identity of the bait proteins, the number of biological replicates, and the false discovery rates (FDR) used for both protein and peptide identification in our repeated IP-MS assays. These details are also reflected in the **modified figure legend (Fig. 4)** and **Supplementary Table 1**.

5. Details on cross-link identification needs to be provided, including the number of cross-linked spectrum matches, unique cross-linked peptides, unique K-K linkages, data reproducibility and FDR.

We thank the reviewer and have now included the requested details in the Materials and Methods and in Table S3. These additions clarify the scope and reliability of the cross-linking MS dataset.

6. All mass spec raw data (IP-MS, XL-MS, HDX) need to be deposited in public repository.

Add data has been uploaded to ProteomeXchange Consortium via the PRIDE and details are provided in Data Availability section (**lines 1227-1230**). The HDX data was from NMR spectroscopy and is not deposited.

Response to Reviewers (responses in red)

Reviewer #1 (Remarks to the Author):

The revised version of the manuscript “Conformational dynamics, RNA binding, and phase separation regulate the multifunctionality of rabies virus P protein” has notably improved, specially from a quantitative perspective.

My concerns, mainly related with the structural characterization, have been properly addressed, and the new version is clearer.

I still have some remarks that I think that will improve the structural part of the manuscript.

Line 470: Dmax is the maximum intramolecular distance and not the particle diameter

We thank the reviewer for this correction. We have amended the text on line 470 to read “maximum intramolecular dimension” to accurately describe Dmax.

Line 483: When describing the conformations of P3 (Fig 5H), one has to imagine a continuous of states going from highly compact to highly extended structures. The use of two states as the authors do is not justified. In general, it is hard to say which protein is more compact when their lengths are different. In my opinion, P3 seems more compact than P1 if one compares the experimental histograms with these of the pools. The clearest signal derived from SAXS data is that P3-D289N is significantly more compact than the other P3 constructs (wt and KRm).

Thank you for this comment and we agree. We have removed the text “two distinct states and on line 483-484 changed the text to “...highlighting the fluctuation between closed and open states where, in the closed states...” to emphasise the heterogeneity of states.

My main criticism with the new version refers to the overinterpretation of the AF models. These models can be used to suggest the presence of interactions between domains, but not to explain some of the cross-links (but not others). AF is very bad in predicting IDRs and, in any case, it represents proteins as a single conformation... but SAXS unambiguous tells the authors that P1 and P3 must be represented by very conformationally heterogenous ensembles. So, I would be very cautious in interpreting the overall architecture of these models when we know that they are highly dynamic.

We agree with the reviewer and fully appreciate this comment. We have removed tying the cross-links to the AF model and now comment on how the AF model suggests the presence of interactions. We have rewritten this section (lines 516 to 525) to remove over-interpretation against the XL-MS data. New/edited text is highlighted to read:

These models propose for P1 that IDR1 interacts extensively with the DD and CTD. The CTD and DD of the P3 model maintain an interaction with the IDR1; however, as IDR1 is truncated, this is consistent with a weaker interaction. An important difference in the models is that for P1 the CTD is oriented whereby the positively charged face (where Lys214 and Arg260 are located) points inwards to the overall structure restricting access to these residues (Figure S10A) whereas in the P3 model, this positively charged face of the CTD points outwards and is exposed (Figure S10B). Nevertheless, it is important to note that these static models of P1 and P3 do not reflect the conformational heterogeneity observed in the SAXS data. Additionally, most predicted cross-links of the XL-MS analysis are not observed in these models, consistent with the low confidence of the modelling of IDR1 and IDR2.

In response, Figure S10 and its legend has been edited.

Reviewer #2 (Remarks to the Author):

I read the revised manuscript and the rebuttal letter addressing the comments. I am satisfied by the responses and with the modifications that the authors made to the manuscript. I do not have additional issues to raise. In my opinion the manuscript can be accepted for publication in Nature Communications.

We thank the reviewer for their positive evaluation and supportive comments.

Reviewer #3 (Remarks to the Author):

The revised manuscript adequately addresses all issues i raised in my former review.

We thank the reviewer for their positive feedback and are pleased that our revisions have addressed all of their concerns.

Reviewer #4 (Remarks to the Author):

While the authors have addressed most of my previous concerns, a few remaining issues regarding result interpretation and data representation require further clarification.

1. Details on cross-link identification including the number of cross-linked spectrum matches, unique cross-linked peptides, unique K-K linkages, and data reproducibility and are still missing. This information should be included in the text.

We thank the reviewer for this helpful suggestion. We have added cross-linked spectrum matches (CSMs) for each identified peptide to Supplementary Table S3 in a separate column, indicating unique peptides with unique Lys–Lys cross-links as described in the table heading.

As suggested, we have also included the abundance values for each cross-link detected in all four sample replicates of P1, P3, P3-D289N, and P3-KRm, along with the median abundance and CV(%), to demonstrate reproducibility.

2. It is unclear how cross-linked peptides were quantified. Since there is no isotope label involved, how the label-free quantitation was performed? The data-dependent acquisition method described in the text cannot be used for accurate label-free quantitation. How were the quantitative values normalized?

We appreciate the reviewer's comment. In response, the precursor intensities for Xlink peptides are used for label-free quantitation (lines 1120-1124), which is different to MaxQuant-based peptide-only level quantitation (lines 974-976). This form of label-free quantitation has been reported previously for cross-linked peptide quantitation, and involves detecting all isotopic peaks in a given dataset that are associated with the cross-linked peptide, the cross-linked spectrum match (CSM). The CSMs are then aggregated to quantification values for the cross-links using the same rules that are applied for peptides. This is a feature of the Minora feature detection, feature mapping, and precursor ion quantifier nodes (Top N algorithm used, N=3) coupled with the XlinkX node in Proteome Discoverer software (Thermo Fisher). As this was a single protein, quantitation was carried out using only the unique cross-linked peptide; therefore we did not perform any peptide-level normalization. We have included this in the revised manuscript (lines 1120-1124):

"...quantifier nodes (Top N algorithm used, N=3) in Proteome Discoverer. As these samples comprised single purified proteins of the same concentration no normalization or scaling was carried out."

We note similar protocols in:

Fasci, D., van Ingen, H., Scheltema, R. A., & Heck, A. J. R. (2018). Histone Interaction Landscapes Visualized by Crosslinking Mass Spectrometry in Intact Cell Nuclei. *Molecular & cellular proteomics : MCP*, 17(10), 2018–33. <https://doi.org/10.1074/mcp.RA118.000924>

Burnap, S. A., Ortega-Prieto, A. M., Jimenez-Guardeño, J. M., Ali, H., Takov, K., Fish, M., Shankar-Hari, M., Giacca, M., Malim, M. H., & Mayr, M. (2023). Cross-Linking Mass Spectrometry Uncovers Interactions Between High-Density Lipoproteins and the SARS-CoV-2 Spike Glycoprotein. *Molecular & cellular proteomics : MCP*, 22(8), 100600. <https://doi.org/10.1016/j.mcpro.2023.100600>.

3. The representation of the qXL data in Figure 6 is somewhat confusing. Quantitation should be shown for cross-linked residue pairs rather than individual residues. Additionally, the color annotations should be clearly explained, and each panel has to have labeled axes.

Figure 6 has been revised. The left-hand panels now show quantitation for cross-linked residue pairs (>2-fold abundance) rather than individual residues. Colour coding is explained on the figure and in the figure legend, and all axes are now labelled.

4. Each supplementary table should have clear and concise table titles. Details on protein

identification are still missing in Table S1. Protein identification details including the number of identified and quantified peptides, identification scores, sequence coverage need to be included.

All Supplementary Tables now have clear and concise titles. In Supplementary Table S1, we have added protein identification details, including the number of identified and quantified peptides, identification scores, and sequence coverage.

5. In Table S3, cells containing '#DIV/0!' or '0' should be replaced with blank spaces or a dash ('-'), as no quantifiable cross-links were present in these cells. Although the authors stated that four biological replicates were performed, only a single abundance value is reported for each cross-link. The experimental results from all replicates should be included and clearly described in the table.

We appreciate the reviewer's comment. We have updated Supplementary Table S3 by replacing '#DIV/0!' and '0' entries with blank spaces to indicate no quantifiable cross-links. Cross-linked spectrum matches (CSMs) for each identified peptide are now included in a separate column. In addition, the abundance values for each cross-link detected in all four sample replicates of P1, P3, P3-D289N, and P3-KRm are now reported in the table.

6. Although the authors stated that cross-links for P1 and P3 support the predicted models in Figure 10. However, the $C\alpha$ - $C\alpha$ distances between cross-linked residues were not shown. These values should be included to illustrate whether the cross-links correlate well with the models. This information should be described in the text.

In line with Reviewer #1's comment regarding the potential overinterpretation of AF models, we have removed discussion of specific cross-links (whether present or absent) in relation to the models. The text now focuses solely on describing potential interactions between domains/regions suggested by the models. Figure S10 and its legend have been revised accordingly to remove cross-link-based interpretation.